# Recurrent innovation of protein-protein interactions in the *Drosophila* piRNA pathway

Sebastian Riedelbauch[1], Sarah Masser [ID][2,3], Sandra Fasching[2], Sung-Ya Lin [ID][4], Harpreet Kaur Salgania[5], Mie Aarup[1], Anja Ebert [ID][1], Mandy Jeske [ID][5], Mia T Levine [ID][4], Ulrich Stelzl [ID][2,3,6] & Peter Andersen [ID][1✉]

## Abstract

**Despite being essential for fertility, genome-defense-pathway genes often evolve rapidly. However, little is known about the molecular basis of this adaptation. Here, we characterized the evolution of a protein interaction network within the PIWI-interacting small RNA (piRNA) genome-defense pathway in *Drosophila* at unprecedented scale and evolutionary resolution. We uncovered the pervasive rapid evolution of a protein interaction network anchored at the heterochromatin protein 1 (HP1) paralog Rhino. Through cross-species high-throughput yeast-two-hybrid screening, we identified three distinct evolutionary protein interaction trajectories across ~40 million years of *Drosophila* evolution. While several protein interactions are fully conserved, indicating functional conservation despite rapid amino acid-sequence change, other interactions are preserved through coevolution and were detected only between proteins within or from closely related species. We also identified species-restricted protein interactions, revealing insight into the mechanistic diversity and ongoing molecular innovation in Drosophila piRNA production. In sum, our analyses reveal principles of interaction evolution in an adaptively evolving protein–protein interaction network, and support intermolecular interaction innovation as a central molecular mechanism of evolutionary adaptation in protein-coding genes.**

**Keywords** Genome Defense; piRNA Pathway; Protein-coding Gene Adaptation; Protein–protein Interactions; Rapid Evolution
**Subject Categories** Chromatin, Transcription & Genomics; Evolution & Ecology

## Introduction

Adaptive phenotypic evolution is driven by the positive selection of beneficial mutations. Notably, the patterns of adaptive evolution across protein-coding genes provide strong predictions about the molecular processes driving such innovation. Specifically, amino acids located at the surface of folded protein domains—and thus available for intermolecular interactions—evolve faster than buried residues not exposed to the surrounding solvent (Moutinho et al, 2019). Consistently, amino acids in protein termini and intrinsically disordered regions, which are often solvent-exposed, evolve faster than central and ordered regions, respectively, and are frequently targeted by adaptive evolution (Afanasyeva et al, 2018; Moutinho et al, 2019; Bricout et al, 2023; Pare et al, 2024). Furthermore, a large-scale comparison of protein architecture, functional annotation, and sequence evolution suggests that adaptive protein evolution is mainly driven by intermolecular interactions, such as protein–protein interactions (Moutinho et al, 2019; Peng et al, 2022). A few experimental studies have examined the evolution of protein–protein interactions between yeast species (Das et al, 2013) or across vast evolutionary time between yeast and humans (Zhong et al, 2016). However, we lack studies with the stepwise granularity in evolutionary time needed to connect to the molecular innovation step in the studied proteins. Specifically, investigations comparing only two species capture only one possible evolutionary trajectory. Other evolutionary outcomes are left undiscovered, limiting our grasp of the diverse mechanisms of sustaining an essential pathway under evolutionary pressure to change. Therefore, large-scale examination of protein–protein interactions in adaptively evolving protein interaction networks remains outstanding.

The PIWI-interacting RNA (piRNA) pathway is a major transposable element (TE) silencing mechanism in the gonads of most animals. Here, PIWI-clade Argonaute proteins guided by their bound piRNAs induce transcriptional and post-transcriptional silencing of target transposable elements (Lau et al, 2006; Watanabe et al, 2006; Aravin et al, 2007; Girard et al, 2006; Grivna et al, 2006; Brennecke et al, 2007). While the piRNA pathway is present in most animals, functional diversification of the piRNA pathway has taken place in the recent evolutionary history of multiple animal species, including several *Drosophila* species (Palmer et al, 2018) and teleost fish (Yi et al, 2014). In *Drosophila*, key genes of the piRNA pathway evolve rapidly under positive selection (Parhad and Theurkauf, 2019), including genes involved in piRNA precursor synthesis (Vermaak et al, 2005), processing (Kolaczkowski et al, 2011; Lee and Langley, 2012), and piRNA-mediated silencing (Obbard et al, 2009a; Kolaczkowski et al, 2011;

[1]Department of Molecular Biology and Genetics, Aarhus University, 8000 Aarhus C, Denmark. [2]Institute of Pharmaceutical Sciences, Pharmaceutical Chemistry, University of Graz, Graz, Austria. [3]BioTechMed-Graz, Graz, Austria. [4]Department of Biology, Epigenetics Institute, University of Pennsylvania, Philadelphia, PA, USA. [5]Heidelberg University Biochemistry Center (BZH), 69120 Heidelberg, Germany. [6]Field of Excellence BioHealth—University of Graz, Graz, Austria. ✉E-mail: pra@mbg.au.dk

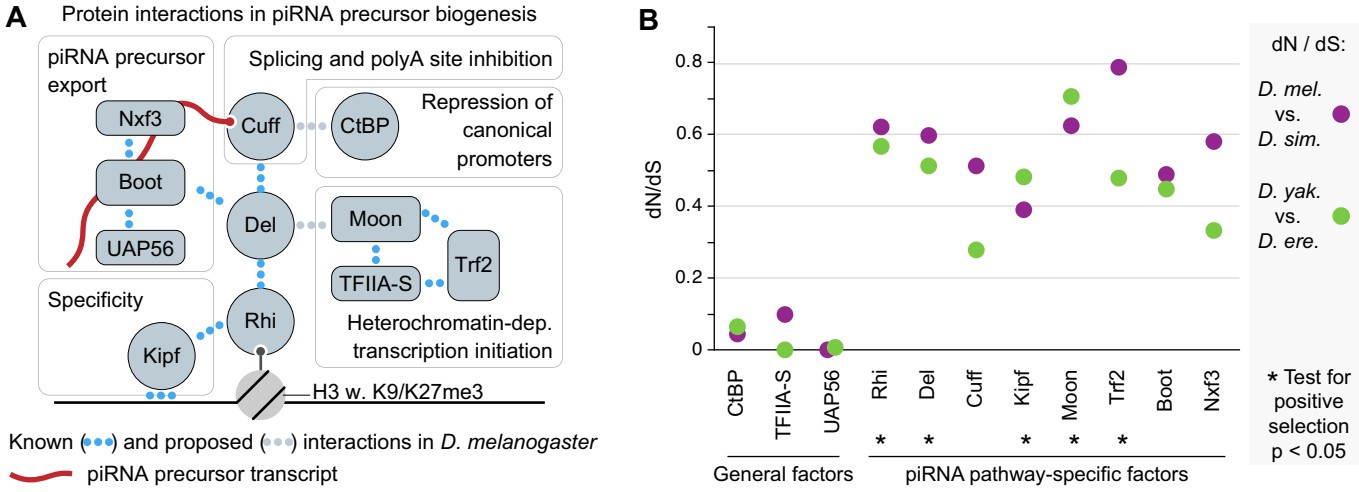

**Figure 1. Rapid evolution of the piRNA biogenesis protein network.**

(A) Schematic showing protein interactions amongst factors enabling germline piRNA precursor biogenesis. Interactions validated to be direct are shown in blue dots, while proposed interactions are in gray. (B) Plot showing dN/dS ratios of genes encoding the proteins shown in (A). Ratios were calculated between *D. melanogaster* and *D. simulans* (purple) and *D. yakuba* and *D. erecte* (green). Genes with significant (*P* < 0.05) CODEML results from PAML suggesting positive selection are indicated by black stars. Source data are available online for this figure.

Lee and Langley, 2012; Simkin et al, 2013). Given the experimental tractability of the *D. melanogaster* model organism, the *Drosophila* piRNA pathway represents an attractive model system for investigating the molecular mechanistic diversification resulting from rapid evolution within animal protein interaction networks.

Among the fastest-evolving piRNA pathway genes in *D. melanogaster* are several involved in germline piRNA precursor biogenesis (Vermaak et al, 2005; Blumenstiel et al, 2016). Most germline piRNAs in *Drosophila* are produced from heterochromatic loci and, therefore, require a dedicated transcription and export machinery for proper expression. At the base of this mechanism is the HP1 homolog protein, Rhino (*rhi*) (Klattenhoff et al, 2009), which is recruited to piRNA source loci through binding to heterochromatic Histone 3 trimethyl-modifications at Lysine 9 (H3K9me3) and Lysine 27 (H3K27me3) (Mohn et al, 2014; Le Thomas et al, 2014; Akkouche et al, 2024) as well as the sequence-specific zinc finger protein, Kipferl (*Kipf*) (Baumgartner et al, 2022). Via the adapter protein Deadlock (Del), Rhino, in turn, recruits a network of effector proteins that enable the production of piRNA precursors from the heterochromatic piRNA source loci (Fig. 1A). First, an alternative basal transcription factor complex consisting of the Transcription factor-IIA (TFIIA) homolog, Moonshiner (Moon), TFIIA-S and TATA box binding protein-related factor 2 (Trf2) is recruited to initiate heterochromatin-dependent piRNA precursor transcription (Andersen et al, 2017). Canonical activation of transposon promoters is thought to be suppressed by the C-terminal Binding Protein (CtBP) transcriptional repressor (Parhad et al, 2019). Next, co-transcriptional splicing, as well as cleavage and polyadenylation, is suppressed via recruitment of the DXO-like protein Cutoff (Cuff) (Pane et al, 2011; Mohn et al, 2014; Zhang et al, 2014; Chen et al, 2016). Finally, piRNA precursor export is mediated by an alternative RNA export complex containing Nuclear export factor 3 (Nxf3), NTF2-related export protein 1 (Nxt1), and Bootlegger (Boot), which further

connects to the general mRNA export complex THO via the UAP56 protein (Zhang et al, 2012; Hur et al, 2016; Kneuss et al, 2019; ElMaghraby et al, 2019). While germline piRNA precursor production is essential for transposon control and female fertility in *D. melanogaster*, several observations suggest that many involved proteins undergo rapid adaptation. For example, hybrid crosses between the closely related species *D. melanogaster* and *D. simulans* show elevated TE activity and reduced germline piRNA precursor production, indicating that the sequence divergence in piRNA precursor biogenesis genes has resulted in a functional divergence between the two species (Kelleher et al, 2012). Furthermore, the *D. simulans* orthologs of Rhino, Deadlock, and Cutoff proteins cannot rescue the respective ortholog mutants in *D. melanogaster*, likely due to divergence in the protein–protein interaction surfaces connecting Rhino, Deadlock, and Cutoff (Parhad et al, 2017; Yu et al, 2018; Parhad et al, 2019). These findings suggest that recurrent molecular innovation, likely at the level of protein–protein interactions, is required to maintain efficient germline transposon silencing in *Drosophila*. However, how adaptive evolution has diversified the proteins but preserved the functional output of piRNA biogenesis remains unknown.

Here, we comprehensively examine the evolution of protein–protein interactions within the piRNA precursor biogenesis pathway in five *Drosophila* species, representing spans of 2–40 million years of evolution. To do so, we performed high-throughput yeast-two-hybrid (Y2H) matrix interaction screening with 11 proteins within and between species, complemented by molecular evolution analysis, *Drosophila* cell culture assays, and in vivo cross-species co-immunoprecipitations (co-IPs) from ovaries. We find that the overall interaction network structure of piRNA precursor biogenesis proteins, despite pervasive signatures of rapid, adaptive evolution, is conserved across the investigated species. However, our data also reveals high evolutionary heterogeneity between individual protein interactions between piRNA precursor

biogenesis factors, including conserved interactions (e.g., Del–Cuff), protein–protein coevolution (e.g., Del–Boot), as well as species-restricted innovation in the protein interaction network by "rewiring" that alters how transcription factors acting at germline piRNA source loci are recruited. Thus, by simply asking the central question of protein–protein interaction compatibility amongst different species, this study shows how rapid evolution has shaped a protein network under positive selection and reveals substantial mechanistic diversity in piRNA precursor production between even closely related species. We speculate that such biochemical innovation reshapes the expressed piRNA population to counteract threats to genome integrity.

# Results

## Rapid evolution and positive selection in genes mediating piRNA precursor biogenesis

Evidence of positive selection or patterns of rapid evolution has been reported for several genes in the *Drosophila* piRNA pathway (Parhad and Theurkauf, 2019). However, a broader coherent molecular evolution analysis, including more recently discovered factors, is missing for a comprehensive understanding of the evolutionary history of the piRNA pathway. To fill this gap, we performed molecular evolution analyses of factors required for germline piRNA precursor biogenesis in *D. melanogaster* (Fig. 1A). This piRNA precursor production relies on highly specialized piRNA pathway genes as well as deeply conserved genes that serve general functions in RNA polymerase II-based gene expression, including TFIIA-S, CtBP, and UAP56. Deeply conserved proteins rarely accumulate amino acid-altering (nonsynonymous) mutation, providing a within-pathway control group. We compared coding sequences of two pairs of closely related Drosophila species pairs (*D. melanogaster* and *D. simulans* as well as *D. yakuba* and *D. erecta*). The dN/dS (ratio of nonsynonymous to synonymous divergence) estimates for all tested piRNA pathway-specific factors were notably elevated compared to the three genes also serving more general roles in gene expression (Fig. 1B). Of note, for Trf2, which functions both in the piRNA pathway (Andersen et al, 2017) and canonical gene expression (Wang et al, 2014; Serebreni et al, 2023), we observed high sequence divergence concentrated in the long isoform, for which the function remains unresolved. Rapid sequence divergence can be explained by either genetic drift (lack of functional constraint) or by adaptive evolution (positive selection). To determine the extent of adaptive evolution within the germline piRNA precursor biogenesis network, we performed likelihood ratio tests of positive selection using the site-specific models in the CODEML program in PAML. We focused on the *melanogaster* species group, spanning ~25 million years of divergence (Russo et al, 1995; Obbard et al, 2012). Comparison of the null model M7 to the alternative model M8 suggested a history of adaptive evolution for Rhino, Deadlock, Kipferl, Moonshiner, and Trf2 (Fig. 1B; Appendix Fig. S1). We also conducted the less sensitive but more conservative model comparison of M8a and M8, which identified Rhino, Kipferl, Moonshiner, and Trf2, but not Deadlock, as positively selected genes (Fig. 1B; Appendix Fig. S1). In sum, we extend previous observations of rapid evolution within the piRNA pathway to uncover pervasive rapid, adaptive evolution in genes

involved in germline piRNA precursor production. These observations suggest that recurrent functional innovation within the essential protein interaction network shapes *Drosophila* piRNA biogenesis.

## A systematic Y2H screen defines the protein interaction network evolution of piRNA precursor biogenesis

Germline piRNA precursor biogenesis is facilitated by protein–protein interactions that lead to the recruitment of effectors (Fig. 1A), driving transcription, co-transcriptional processing, and piRNA precursor export (Mohn et al, 2014; Zhang et al, 2014; Andersen et al, 2017; ElMaghraby et al, 2019; Kneuss et al, 2019). Recent studies have uncovered incompatibilities amongst the Rhino, Deadlock, and Cutoff factors from the closely related *D. melanogaster* and *D. simulans* species (Parhad et al, 2017, 2019). Given the recent predictions that adaptive evolution in protein-coding genes is mainly driven by intermolecular interactions rather than innovation of, e.g., protein stability or folding (Moutinho et al, 2019; Peng et al, 2022), we hypothesized that the functional innovation suggested by our evolutionary analyses has occurred at the level of protein–protein interactions. To test this hypothesis, we performed a cross-species, all-versus-all yeast-two-hybrid (Y2H) matrix screen, a method previously used to study host-pathogen relationships (reviewed in (Shah et al, 2015; Goodacre et al, 2020)), protein interactome essentiality (Ghadie and Xia, 2019), and subcomplex contacts in the nuclear pore complex (Apelt et al, 2016). Here, we screened 11 proteins involved in germline piRNA precursor biogenesis (Fig. 1A) from five different *Drosophila* species (*D. melanogaster*, *D. simulans*, *D. erecta*, *D. persimilis*, and *D. virilis*), representing different evolutionary distances across 40 million years of divergence (Fig. 2A). As Kipferl is absent in *D. persimilis* and *D. virilis* (Baumgartner et al, 2022), this resulted in a set of 54 proteins from five species (Table EV1; Appendix Figs. S2–S5) allowing a systematic assessment of 308 intra- and 1123 inter-species unique protein pairs for interaction.

Y2H experiments are prone to false negative results due to specific tagging, as fusions might interfere with folding or mask binding surfaces. To reduce the potential false negative rate (Venkatesan et al, 2009; Stelzl, 2014), we cloned all coding sequences as bait and prey, each fused C- and N-terminally with a DNA-binding (DBD) or a transcription activation domain (AD), respectively. Therefore, every protein is represented as four individual constructs, and every protein interaction is tested in eight combinations using different constructs (Woodsmith et al, 2017) (Figs. 2A and EV1A,B). Bait proteins that did not show any interactions or, conversely, resulted in a set of well-sampled protein–protein interactions were screened twice. The other bait was screened three times, with three or four technical replicates each (Fig. EV1C–F). Excluding 10 auto-active bait and two auto-active prey constructs from the analyses, we examined a total of 89,088 individual pairs. For scoring Y2H interactions, we applied thresholds for colony size and technical as well as biological replication (Appendix Fig. S6). Here, each unique configuration of a protein–protein interaction can lead to a maximum score of one. After combining all eight potential configurations for a protein pair, the sum of the scores can reach a maximum of eight. We then benchmarked the success of the screen by assessing protein–protein interactions previously described between piRNA precursor

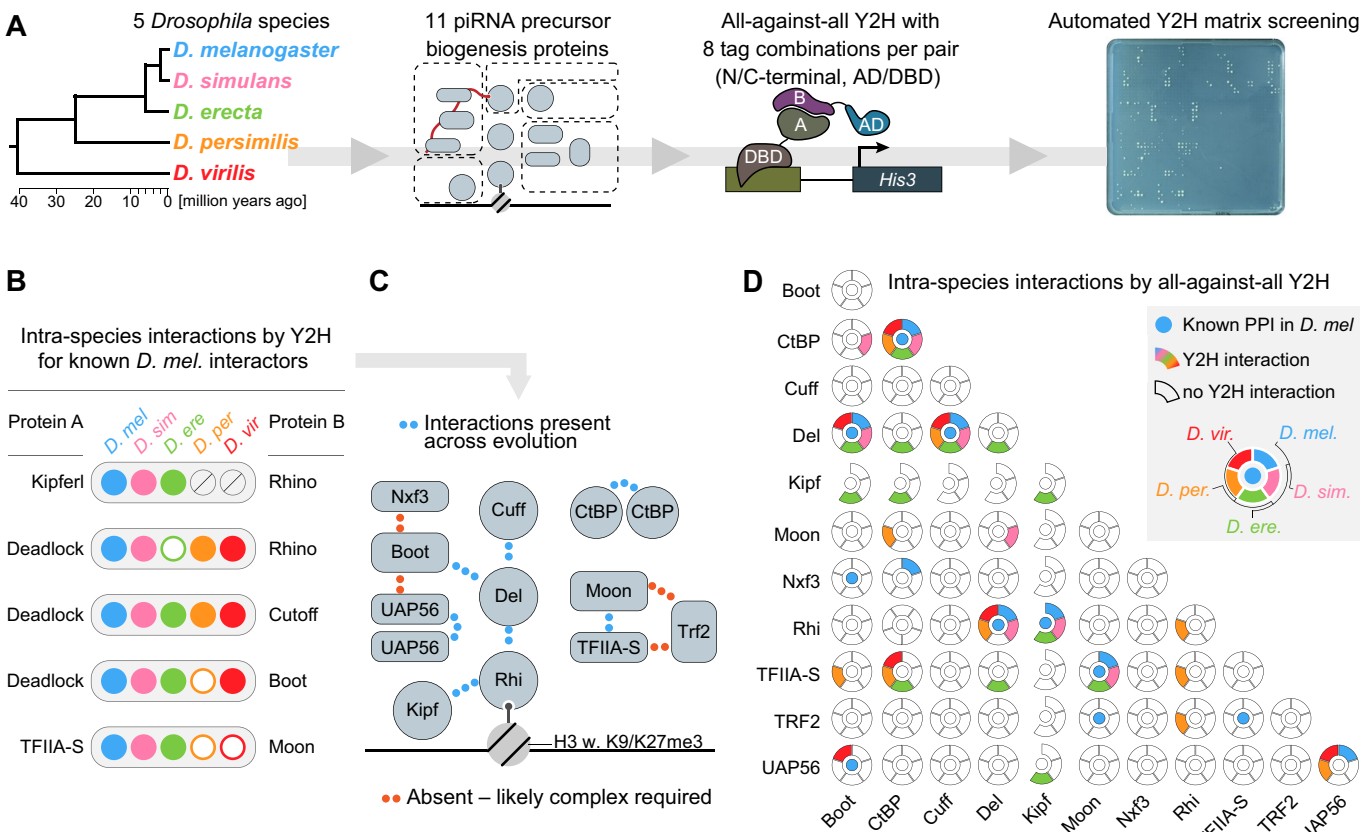

**Figure 2. A yeast-two-hybrid screen uncovers the evolutionary biochemistry of piRNA biogenesis.**

(A) Schematic showing the experimental strategy of the yeast-two-hybrid screen. The major protein factors in germline piRNA precursor production are cloned from five *Drosophila* species spanning 40 million years of evolution and subsequently tested in all-against-all yeast-two-hybrid matrix screens. AD activation domain, DBD DNA-binding domain. See also Fig. EV1. The species color code shown in the phylogenetic tree is the same throughout the paper. (B) Schematic summary of yeast-two hybrid results for interactions known from *D. melanogaster* amongst piRNA biogenesis factors within the same species (intra-species interactions). Full circles indicate detected interactions above the replication score threshold of 0.75, while empty circles indicate that interaction was not detected or the score was below the threshold. Gray crossed-out circles for Kipferl denote the absence of the gene in those species. (C) Schematic summary of the conserved interactions identified across the tested species phylogeny. (D) Schematic summary of yeast-two hybrid results for all tested intra-species interactions amongst piRNA biogenesis factors. Central blue dot: known interaction from *D. melanogaster*. Surrounding circle blocks indicate whether the interaction was detected (full block) or not (empty block). The color scheme follows the species annotation from (A). Source data are available online for this figure.

biogenesis factors in *D. melanogaster*. Using a conservative Y2H replication score cutoff of 0.75, we report 199 interacting protein pairs, 46 intra- and 153 inter-species interactions (Appendix Fig. S6). The five most central protein interactions known from studies of the network in *D. melanogaster* were confirmed by Y2H (Fig. 2B, blue dots). The four previously described interactions that were not detected by Y2H (Boot–Nxf3, Boot–UAP56, Trf2–Moon, and Trf2–TFIIA-S) (Fig. 2B,C) are all part of multimeric complexes in the form of either Nxf3–Nxt1–Bootlegger–UAP56 (Kneuss et al, 2019; ElMaghraby et al, 2019) or Trf2–TFIIA-S–Moonshiner (Andersen et al, 2017). Co-IP and Y2H approaches are known to yield highly complementary interaction information (Stelzl, 2014). Only about 30% of the protein pairs in AP-MS co-IP data reflect direct physical interactions only (Kunowska and Stelzl, 2022). In contrast, Y2H assays by design specifically probe direct binary protein–protein interactions, and interactions that depend on the formation of multimeric complexes will therefore not be detected, likely explaining the absence of the Boot–Nxf3, Boot–UAP56, Trf2–Moon, and Trf2–TFIIA-S in our data. In addition, the

previously described ability of CtBP and UAP56 to homodimerize (Bhambhani et al, 2011; Zhao et al, 2004) was supported by Y2H signal in tests where the same protein was tagged as bait and prey (Fig. EV2). We, therefore, conclude that our Y2H screen recapitulated known pairwise interactions within the germline piRNA biogenesis pathway.

Not only are the vital *Drosophila* piRNA pathway genes evolving rapidly, but some proteins found to be essential for the piRNA pathway in *D. melanogaster* are dispensable in closely related species (Chary and Hayashi, 2023), revealing a remarkable evolutionary plasticity within the *Drosophila* piRNA pathway. We therefore investigated the degree of conservation of the core interaction network of germline piRNA biogenesis factors by comparing intra-species protein interactions across the investigated *Drosophila* species. Our Y2H assays showed that the central interactions connecting Deadlock to Rhino, Cutoff, and Bootlegger are detected in at least three of the four tested non-*melanogaster* species (Figs. 2B,C and EV2). Similarly, we observed interaction between Rhino and Kipferl in all species harboring the *kipferl* gene.

Furthermore, we did not find evidence for the proposed interaction between Cutoff and CtBP (Parhad et al, 2019), and only the *D. simulans* orthologs supported the proposed interaction between Moonshiner and Deadlock (Andersen et al, 2017) (Figs. 2B–D and EV2). These results could reflect that (i) the Cutoff–CtBP and Moonshiner–Deadlock interactions are technically challenging to detect by Y2H, (ii) CtBP and Moonshiner are not involved in piRNA precursor biogenesis in some *Drosophila* species, or (iii) unknown and potentially evolutionarily labile interactions connect CtBP and Moonshiner to Rhino-dependent piRNA source loci in different *Drosophila* species. In sum, we uncovered an overall conserved interaction network structure amongst *Drosophila* germline piRNA biogenesis factors—with notable exceptions indicating evolutionarily labile recruitment of the CtBP and Moonshiner transcription factors.

## Cross-species analysis reveals diverse protein interaction evolution within the piRNA pathway

The conservation of a core protein interaction network within the piRNA pathway can be explained by at least two evolutionary trajectories: (i) protein interaction site conservation and (ii) protein interaction coevolution, whereby compensatory mutations evolve that restore protein–protein interaction following sequence changes that otherwise diminish it. Interestingly, such interaction site coevolution has previously been suggested for interactions between Rhino, Deadlock, and Cutoff based on observed species incompatibilities between *D. melanogaster* and *D. simulans* (Parhad et al, 2017, 2019). We, therefore, looked for signatures of either interaction site conservation or coevolution in our cross-species Y2H data representing evolutionary distances from 0 to 40 million years of divergence (Russo et al, 1995; Obbard et al, 2012) (Fig. 3A). We reasoned that conserved interaction surfaces should yield Y2H interaction signals for proteins across the tested species (interspecies interactions), while co-evolving interactions should instead tend to show interaction signals between proteins from the same species (intra-species interactions) or from closely related species. We found that a high fraction of the tested Deadlock–Cutoff and Rhino–Kipferl interactions are detected by Y2H in both intra-species and inter-species interaction tests across all tested evolutionary distances (Figs. 3B and EV3A), compatible with a model of conserved interaction surfaces. Conversely, the interactions of Deadlock–Rhino and Deadlock–Bootlegger showed a higher fraction of intra-species and short-distance inter-species interactions compared to tests between more distantly related orthologs (Fig. 3C,D), indicative of a model of coevolution. Consistent with these findings, coevolution of the Deadlock–Rhino was previously proposed to have taken place based on comparative functional and structural studies across *D. melanogaster* and *D. simulans* (Parhad et al, 2017; Yu et al, 2018). Our Y2H data, however, also indicate functional innovation beyond the conserved interactions in the piRNA precursor biogenesis network, as evident from several species-restricted interactions supported by Y2H interactions detected only within one or two *Drosophila* species. For example, we found that *D. persimilis* Rhino interacts with the transcription factors TFIIA-S and Trf2 (Fig. EV3B,C). Such interaction with TFIIA-S and Trf2 from multiple species indicates that the molecular innovation enabling the interaction has occurred along the lineage leading to the *D. persimilis* Rhino. Similarly, we

observed that Deadlock orthologs from *D. simulans* and *D. erecta* interact with Moonshiner from several species (Fig. 3E). Finally, we also detected interaction between CtBP from most species and the Deadlock orthologs from *D. erecta* as well as *D. virilis* (Fig. 3F). In sum, contrasting the expectation of conserved biochemical functions in essential proteins, the presented Y2H data suggests that individual protein interactions in piRNA precursor biogenesis follow separate evolutionary paths and are characterized by either conserved, co-evolving, or species-restricted patterns.

The protein–protein interactions that enable piRNA precursor production occur in *Drosophila* germline cells in a very different molecular environment from the yeast cells used in Y2H assays. When assayed by confocal microscopy, the investigated proteins accumulate in distinct foci within the ovarian germline nurse cell nuclei (Klattenhoff et al, 2009; Zhang et al, 2012; Mohn et al, 2014; Andersen et al, 2017; Kneuss et al, 2019; ElMaghraby et al, 2019; Baumgartner et al, 2022). These foci represent major piRNA source loci, and the accumulation of the individual proteins depends on the protein–protein interactions that directly or indirectly connect them to Rhino bound to the locus chromatin (Fig. 1A). To probe how well our Y2H data translate to functionally relevant protein interactions in vivo, we generated transgenic *D. melanogaster* flies expressing GFP-tagged Bootlegger, Deadlock or Kipferl specifically in germline cells driven by the *rhino* promoter of *D. melanogaster*, which facilitates transgene expression similar to the endogenous *rhino* gene (Baumgartner et al, 2022; Fig. 3G). To assay protein interactions in their native environment, we first performed immunofluorescence confocal imaging of nurse cell germline nuclei and monitored the co-localization of the transgenic GFP-fusions with endogenous *D. melanogaster* Rhino. Importantly, since the formation of strong Rhino foci relies on the presence of *D. melanogaster* Deadlock (Mohn et al, 2014; Parhad et al, 2017), we performed the localization in wildtype endogenous Deadlock background. Consistent with the Y2H data, we observed that Deadlock from *D. melanogaster*, *D. simulans*, and *D. virilis*, but not from *D. erecta*, colocalizes with endogenous *D. melanogaster* Rhino (Figs. 3H and EV3D). Of note, *D. simulans* Deadlock was also previously observed to co-localize with endogenous Rhino when expressed in *D. melanogaster* (Parhad et al, 2017). Similarly, we observed complete congruence of Y2H data and in vivo protein co-localization for the tested interactions of Rhino–Kipferl (Figs. 3I and EV3E) and Deadlock–Bootlegger (Fig. EV3F, Bootlegger localization to Rhino foci relies on Deadlock). We conclude that patterns of conserved as well as co-evolving protein interactions inferred from our Y2H data are recapitulated by in vivo protein co-localization assays.

## Recurrent protein network rewiring identified by species-restricted protein interactions

In addition to the conserved or co-evolving interactions that place Deadlock and Bootlegger at central molecular hubs in piRNA precursor biogenesis (Figs. 2B,C and 3D), our Y2H data also suggest that both proteins take part in several species-restricted protein–protein interactions. For example, *D. simulans* Bootlegger and *D. erecta*, as well as *D. virilis* Deadlock, showed highly reproducible interaction with CtBP in some species but not others (Figs. 2D, 3F, and EV2) and Deadlock from both *D. simulans* and *D. erecta* interacted with Moonshiner from several species

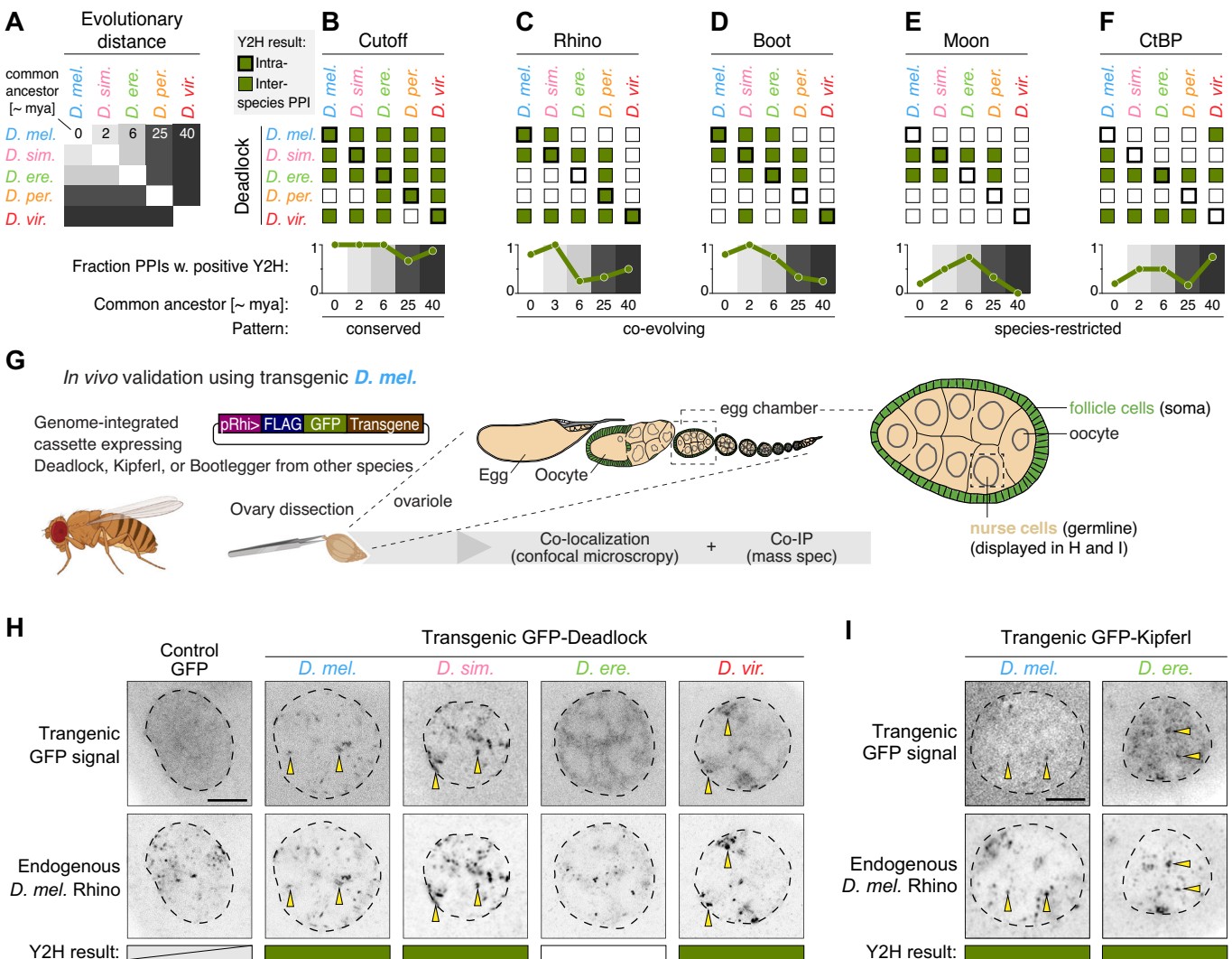

**Figure 3. Protein–protein interaction evolution uncovered by cross-species interaction analysis.**

(A) Diagram indicating the estimated evolutionary distance between the tested Drosophila species. mya million years ago. (B–F) Summary of interactions detected by yeast-two-hybrid between orthologs of Deadlock and Cutoff (B), Rhino (C), Bootlegger (D), Moonshiner (E), and CtBP (F). Intra-species (thick box outline) and inter-species (thin box outline) interactions are shown as filled green boxes. In contrast, empty boxes indicate the absence of interaction detection above the replication score threshold. The plots below each summary matrix show the fraction of positive Y2H interactions split by the evolutionary distance between the source species. The fractions were calculated as the number of positive interactions divided by the number of tested interactions in each evolutionary distance group (see also (A)). Pattern: evolutionary signature of the interaction (see main text for details). (G) Schematic of the transgenic strategy testing interaction between proteins from different species (inter-species) in vivo by co-localization and co-immunoprecipitation analysis. (H, I) Confocal microscopy images showing the localization of endogenous *D. melanogaster* Rhino (anti-Rhi IF) and GFP-tagged transgenic Deadlock (H) or Kipferl (I) from the indicated species. Yellow arrows highlight co-localizing foci of endogenous Rhino IF signal and transgenic GFP-tagged proteins. Dashed line: nuclear border determined by DAPI staining (see Fig. EV3D,E). Scale bars indicate 5 µm. Source data are available online for this figure.

(Figs. 3E and EV2) but not others. To more directly characterize such species-restricted protein–protein interactions in vivo, we next performed co-immunoprecipitation coupled to mass spectrometry analysis (co-IP/MS) using ovary lysates from the transgenic *D. melanogaster* fly lines expressing Bootlegger and Deadlock proteins from various species. Co-IP/MS analysis of C-terminally tagged *D. melanogaster* Bootlegger recapitulated interactions identified in previously published co-IP/MS data, such as with Nxf3 (ElMaghraby et al, 2019) (Fig. EV4A,B). The C-terminally tagged *D. melanogaster* Bootlegger also co-precipitated with all three PIWI-clade Argonaute proteins (Piwi, Aub, Ago3, Fig. EV4A,B),

suggesting direct coupling of piRNA precursor export and cytoplasmic processing to mature piRNAs. Conversely, N-terminal tagging somehow disrupts the strong co-IP association of *D. melanogaster* and *D. simulans* Bootlegger with the piRNA precursor export complex proteins UAP56, Nxt1, and Nxf3 (Figs. 4A–C and EV4A,B). Consistently, nuclear export of N-terminally tagged Bootlegger is disrupted, as evident from the lack of the peri-nuclear foci observed for fully functioning C-terminally tagged Bootlegger (Appendix Fig. S7). This disruption, however, did not perturb the species-restricted interactions detected by Y2H, as co-IP/MS of N-terminally tagged *D. simulans*

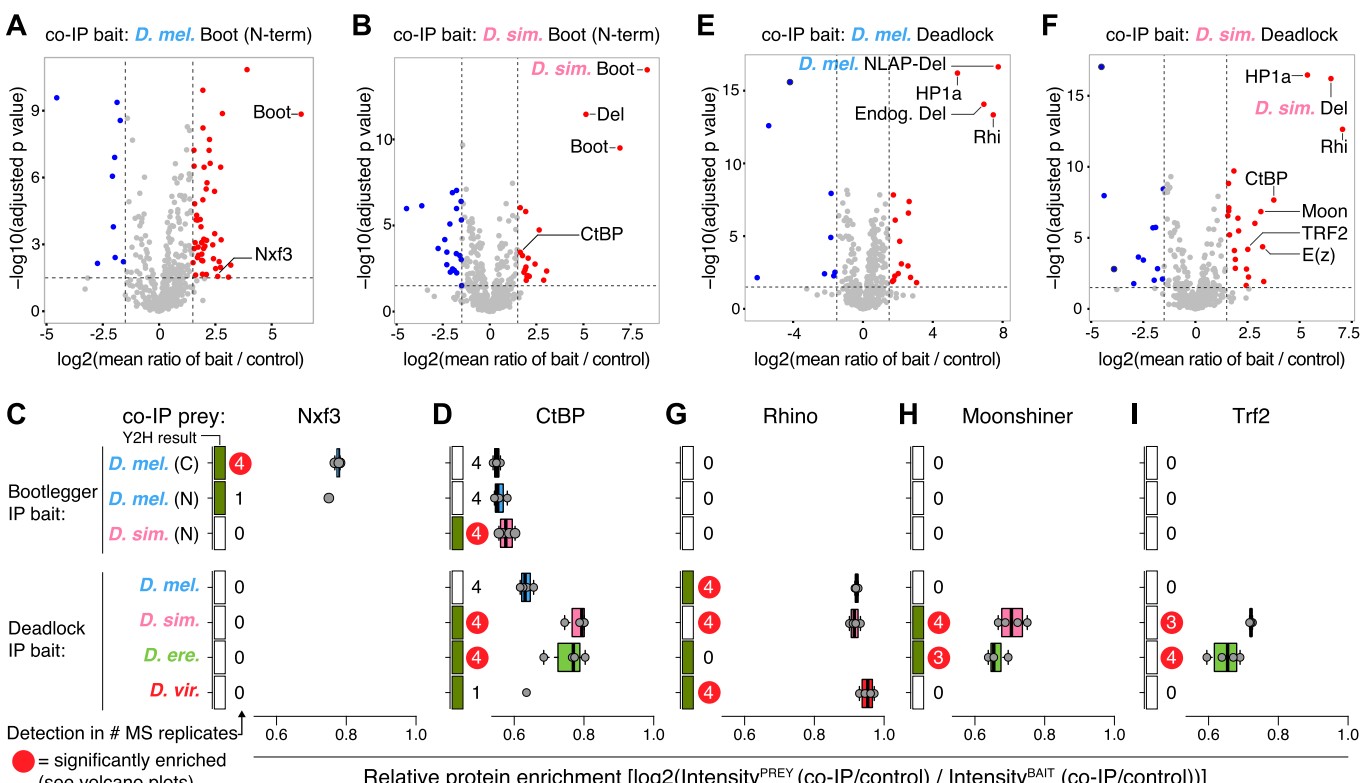

**Figure 4. Cross-species in vivo Co-IP/MS shows the innovation of protein interactions in piRNA biogenesis.**

(A, B) Volcano plots displaying co-IP/mass spectrometry data from the co-IP baits indicated above each plot. The x axes show log2 fold change between bait and control IP averaged over four biological replicates with every two technical replicates. The y axis displays the negative log10 of adjusted P values from t tests for enrichment in bait compared to control co-IP. Red and blue dots represent proteins with more than 1.5-fold change with −log10(adjusted P values) higher than 2. (C, D) Box plots showing co-IP protein binding enrichment relative to GFP-tagged bait binding for the indicated bait and prey proteins. The whiskers denote Tukey's Fences at 1.5 times the interquartile range beyond the first and third quartiles. Rectangles at the left y axis indicate whether the protein interaction was detected by yeast-two-hybrid (filled green) or not (empty white). Numbers denote the number of co-IP biological replicates in which the prey protein was detected. Red circles indicate that the prey protein was determined to be statistically significantly enriched (see A, B, E, F, and Fig. EV4B–D). Gray dots represent enrichment values for individual biological replicates averaged over two technical replicates. (E, F) Volcano plots are similar to (A, B). (G–I) Box plots similar to (C, D). Source data are available online for this figure.

Bootlegger showed prominent co-precipitation of both Deadlock and CtBP (Fig. 4B,D). In sum, we demonstrate the species-restricted robust interactions between *D. simulans* Bootlegger and Deadlock as well as CtBP by both Y2H and inter-species co-IP/MS analysis from transgenic flies.

In the *D. melanogaster* Deadlock co-IP/MS analysis, Rhino and HP1a were co-immunoprecipitated (Fig. 4E). In contrast, Cutoff and Bootlegger were not, despite the strongly supported interaction between these proteins based on Y2H (Figs. 3B,D and EV2). This suggests that tagged Deadlock transgenes may only partially recapitulate endogenous Deadlock interactions. However, consistent with our co-localization analysis (Fig. 3H) and the proposed coevolution model, Rhino protein was identified in co-IP/MS analysis of *D. simulans* and *D. virilis* Deadlock, but not in *D. erecta* Deadlock co-IP/MS (Figs. 4E–G and EV4C,D). In addition, the species-restricted interaction between Deadlock and Moonshiner identified by Y2H (Figs. 3E and Fig. EV2) was supported by pronounced co-precipitation of Moonshiner as well as the Moonshiner binding partner Trf2 (Figs. 4FH,I and EV4C). Given the Y2H-based *D. persimilis*-specific interactions between Rhino and TFIIA-S as well as Trf2 (Fig. EV2), our data collectively indicate

recurrent innovation of the protein–protein interactions leading to RNA Polymerase II recruitment at germline piRNA source loci (see "Discussion").

Surprisingly, we also observed notable co-precipitation of Enhancer of Zeste (E(z)) protein in co-IP/MS analysis of *D. simulans*, *D. erecta*, and *D. virilis* Deadlock (Figs. 4F and EV4A,C,D). E(z) is the catalytic subunit of the Polycomb Repressive Complex 2 (PRC2), which deposits the repressive H3K27me3 mark (Müller et al, 2002; Cao et al, 2002; Czermin et al, 2002). Of note, amongst the seven known PRC2 proteins, only E(z) was observed to be enriched in the Deadlock co-IPs (Fig. EV4A), indicating that E(z) may be recruited to Deadlock alone. H3K27me3 marks were recently shown to mediate Rhino binding at many germline piRNA source loci in *Drosophila* (Akkouche et al, 2024). The identified connection between Deadlock and E(z), therefore, has potential implications for piRNA source locus regulation, as it implies that H3K27me3 may be deposited as a consequence of Rhino–Deadlock association with loci rather than acting upstream of such binding (see "Discussion").

Finally, co-IP/MS analysis uncovered co-precipitation of *D. melanogaster* CtBP with Deadlock from *D. simulans* and *D. erecta*

and—substantially weaker—with Deadlock from *D. melanogaster* but not with Deadlock from *D. virilis* (Figs. 4D–F and EV4C,D). Our Y2H data indeed support direct interaction between CtBP and both *D. simulans* and *D. erecta* Deadlock (Fig. EV2). Furthermore, *D. virilis* Deadlock interaction with CtBP was strongly supported by Y2H data (Fig. EV2) but was not recapitulated by co-IP/MS (Figs. 4D and EV4A,D). Notably, the co-precipitations of CtBP with *D. simulans* and *D. erecta* Deadlock were also accompanied by robust interaction with the Moonshiner–TFIIA-S–Trf2 complex (Fig. 4H,I). Across all tested species, low but consistent Y2H scores supported interaction between TFIIA-S and CtBP (Fig. EV2). We, therefore, speculate that robust Deadlock–Moonshiner interaction may stabilize CtBP interactions with Deadlock through additional CtBP-TFIIA-S interactions.

In sum, our Y2H data, together with in vivo co-localization and co-IP/MS analyses, reveal multiple species-restricted protein interactions amongst the proteins facilitating transcriptional regulation at germline piRNA source loci.

## Short linear motif evolution facilitates mechanistic divergence in CtBP recruitment

To understand which changes in protein sequence may facilitate the observed innovation in protein–protein interactions, we next focused on the rewiring of CtBP interactions. Close inspection of CtBP Y2H interaction data revealed a highly variable pattern of interaction partners across the investigated species (Fig. 5A–G). CtBP dimerization, which is known to facilitate its repressive functions (Bhambhani et al, 2011), was robustly detected within and between all species (Figs. 5A and EV2); however, interactions with TFIIA-S, Deadlock, Nxf3, Kipferl, Moonshiner, and Bootlegger were found in only a few or a single species (Figs. 5B–G and EV2). The CtBP interaction with TFIIA-S was observed across all species but *D. melanogaster*, while only *D. erecta* and *D. virilis* Deadlock interacted with the CtBP proteins. Nxf3, Kipferl Moonshiner, and Bootlegger interaction with CtBPs were specific for *D. melanogaster*, *D. erecta*, *D. persimilis*, and *D. simulans*, respectively (Fig. 5D–G). Importantly, given the high conservation of the CtBP amino acid sequence across the tested species (Appendix Fig. S8), Y2H assays with CtBP from each species serve as further replicates, attesting to the robustness of the identified interaction signals.

To understand the molecular mechanism underlying recurrent CtBP interaction rewiring, we decided to characterize the interaction between *D. simulans* Bootlegger and CtBP (Figs. 4B and 5G). To this end, we used a qualitative, fluorescence-based re-localization protein–protein interaction assay (ReLo) (Pekovic et al, 2023; Kubíková et al, 2023; Salgania et al, 2024a). Plasmids encoding candidate interacting proteins fused to a fluorescent protein and a membrane-associated anchor are co-transfected into *Drosophila* S2R+ cells. Protein–protein interactions are then assayed by co-localization analysis using live cell confocal imaging. To adapt the ReLo assay to nuclear proteins, we fused the proteins with either a mitochondrial outer membrane anchor (Daed$^{TMM}$) or an endoplasmic reticulum (ER) membrane anchor (OST4), thus forcing the fusion proteins to remain in the cytoplasm. Confirming our results from Y2H and co-IP/MS assays, we observed that Bootlegger-mCherry-Daed$^{TMM}$ displays co-localization with ER-anchored OST4-GFP-CtBP protein when the assay was done using

*D. simulans* ortholog. In contrast, this was not observed for the respective proteins from *D. melanogaster* (Fig. 5H,I), thus further supporting a species-restricted interaction between Bootlegger and CtBP in *D. simulans*. Of note, while control OST4-GFP localized to cytoplasmic endoplasmic reticulum-like regions, OST4-GFP-CtBP protein accumulated in ring structures around the nucleus with some cytoplasmic foci (Fig. 5H,I). We speculate that such accumulation of the CtBP fusion protein reflects a strong nuclear localization signal, which traps the proteins at the nuclear envelope due to its simultaneous ER association.

To identify the *D. simulans* Bootlegger interaction surface to CtBP, we generated a series of chimeric expression constructs with systematic domain swaps between *D. melanogaster* and *D. simulans* Bootlegger (Figs. 5J and EV5A). ReLo assays using these constructs revealed that a central 50-amino acid region in *D. simulans* Bootlegger is necessary and sufficient for interaction with CtBP (Fig. EV5A–C). Inspection of this region uncovered a putative PIDLRS short linear motif (Fig. EV5D), previously identified to facilitate interaction between human hCTBP2 protein and the HCP2 polycomb group developmental regulator (Sewalt et al, 1999). To test whether the PIDLRS motif is required for *D. simulans* Bootlegger interaction with CtBP, we mutated either one or both of the amino acids in the motif that differ from the corresponding *D. melanogaster* sequence (I224T, L226P). ReLo assays using *D. simulans* Bootlegger with single or double *D. melanogaster*-like motif mutation showed complete loss of co-localization (Fig. 5J,K), suggesting that a short linear binding motif specific to *D. simulans* Bootlegger is necessary and sufficient for binding to CtBP. Notably, in *D. erecta* Deadlock, for which both Y2H and co-IP/MS data strongly supported robust interaction with CtBP (Figs. 3F and 4B), we also identified a variant CtBP interaction motif (PLDLS) (Fig. EV5E), known to facilitate CtBP interaction in animals (Turner and Crossley, 2001). We demonstrate that the evolution of a short linear binding motif can drive the rewiring of the piRNA pathway protein–protein interaction network in *Drosophila*.

# Discussion

Here, we investigated the biochemical consequences of rapid evolution within 11 piRNA precursor biogenesis genes across five *Drosophila* species spanning an evolutionary distance of 40 million years. Using a well-proven, high-quality Y2H matrix array screening setup, we comprehensively tested the physical and binary protein interactions of the 11 piRNA pathway members of *D. melanogaster*, *D. simulans*, *D. erecta*, *D. virilis*, and *D. persimilis* in a pairwise, all-versus-all screen, yielding both comprehensive within (intra) and cross-species (inter) protein interaction data. Based on a high-quality data set of 199 protein interactions (46 intra and 153 inter-species), we elucidated evolutionary patterns of interaction, including strict conservation, coevolution, and species-restricted interactions. These inferences were further validated and characterized using in vivo co-immunoprecipitation and cell culture assays.

## Innovation of protein–protein interactions underlie rapid evolution in the piRNA pathway

Our work links the adaptive evolution of protein sequence and the underlying innovations in biochemical functionality. According to

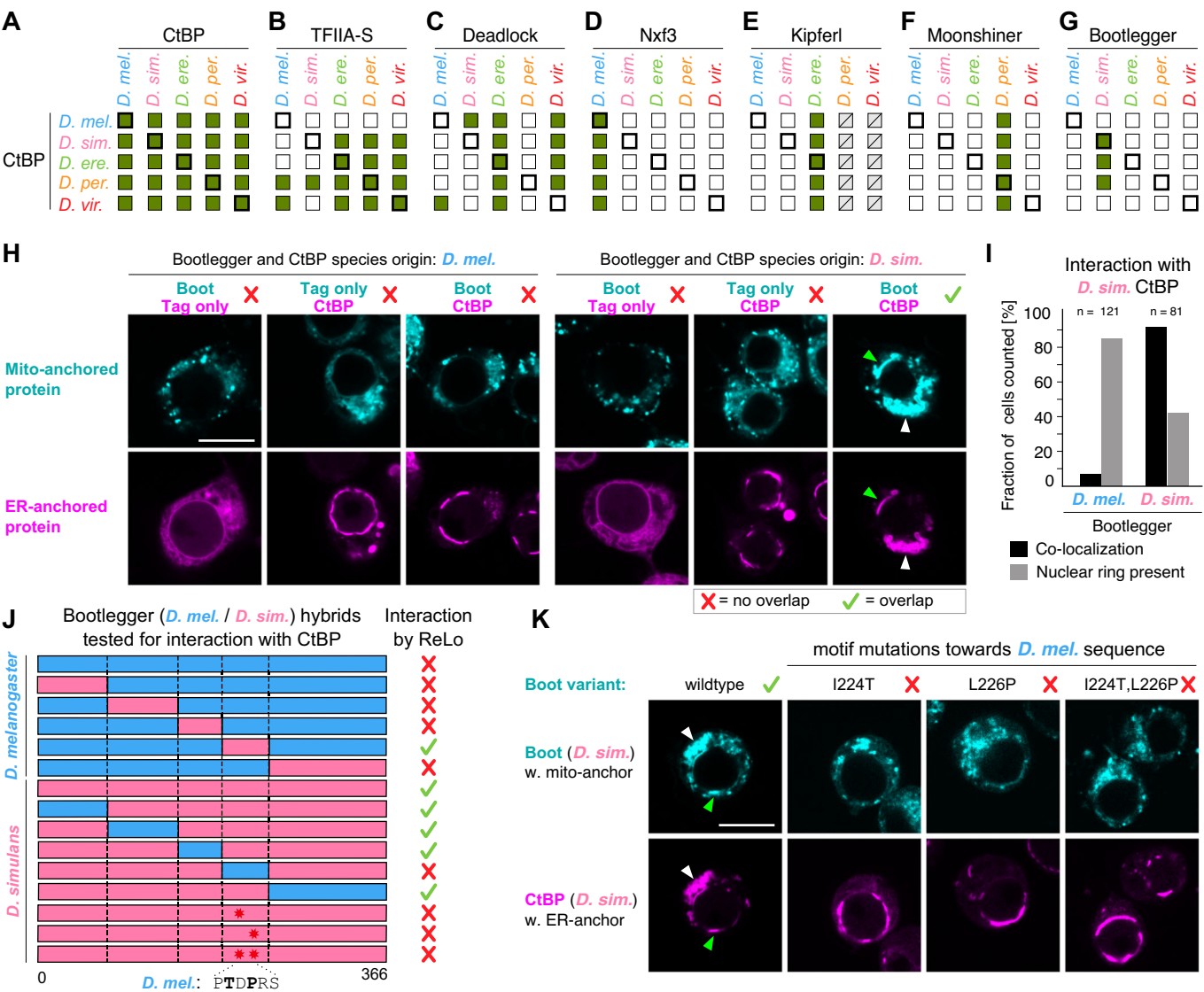

**Figure 5. Recurrent innovation of CtBP interaction with piRNA biogenesis factors.**

(A–G) Summary of interactions detected by yeast-two-hybrid between orthologs of CtBP and CtBP (A), TFIIA-S (B), Deadlock (C), Nxf3 (D), Bootlegger (E), Kipferl (F), and Moonshiner (G). Intra-species (thick box outline) and inter-species (thin box outline) interactions are shown as filled green boxes. In contrast, empty boxes indicate the absence of interaction detection above the replication score threshold. (H) Fluorescence microscopy images of ReLo protein interaction assays testing interaction between CtBP and Bootlegger proteins from *D. melanogaster* (left panel) and *D. simulans* (right panel). Shown are representative cells from >10 images. Green arrows indicate protein accumulation in interrupted ring structures around the nucleus. White arrows show protein accumulation at cytoplasmic sites. Scale bars indicate 8 μm in size. Red crosses and green check marks denote whether protein interaction was concluded or not, respectively, based on the ReLo assays. (I) Bar plot displaying the quantification of ReLo co-localization and nuclear ring-like phenotypes in 202 individually evaluated cells. (J) Schematic representation of domain-swap constructs between the *D. melanogaster* (blue) and *D. simulans* (pink) Bootlegger orthologs with the interaction results indicated to the right by red crosses or green check marks. Black stars indicate the tested interaction site point mutants. (K) Fluorescence microscopy images of ReLo protein interaction assays testing point mutants of *D. simulans* Bootlegger for interaction with CtBP. Displayed as for (H). Source data are available online for this figure.

evolutionary theory, essential cellular pathways should undergo strong negative (purifying) selection because changes in proteins could lead to lethality. However, evolutionary studies have found that genes supporting certain functions, such as immune response (Nielsen et al, 2005; Sackton et al, 2007; Obbard et al, 2009b; Slodkowicz and Goldman, 2020) and defense against viruses and transposons (Obbard et al, 2009a; Enard et al, 2016; Palmer et al, 2018) often evolve rapidly with selection for amino acid-changing variants (positive selection) (Zhang and Yang, 2015; Booker et al, 2017). For example, genes facilitating siRNA- and piRNA-mediated defense against viruses and transposons evolved much faster than genes involved in microRNA-mediated general gene regulation (Obbard et al, 2009a). Yet, the innovation in biochemical functionality in such rapidly evolving, albeit essential genes has been characterized in only a few cases, such as the primate retroviral restriction factor TRIM5α (Sawyer et al, 2005), mouse

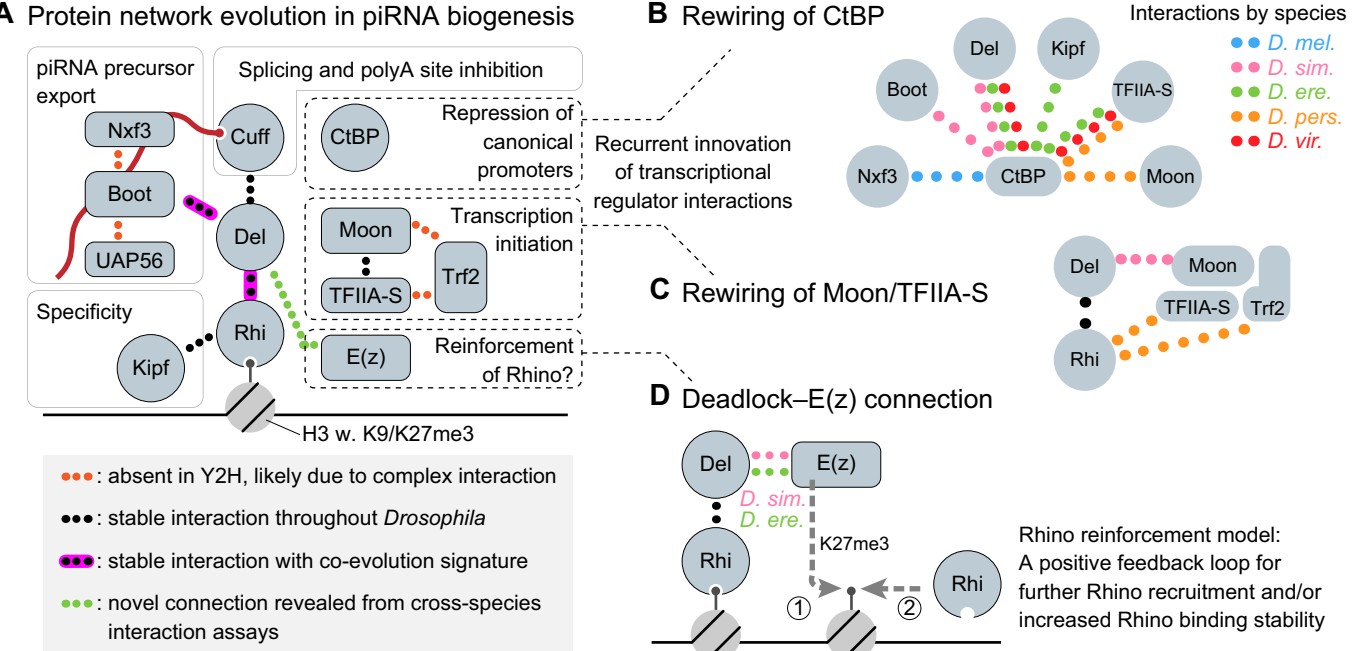

**Figure 6.  Evolutionary biochemistry of the germline piRNA biogenesis factor network.**

(A) Schematic model summarizing the identified evolutionary characteristics for individual protein–protein interactions within the germline piRNA precursor biogenesis network. Dotted lines represent detected protein–protein interactions. See legend for details. (B, C) Schematic models of the observed evolutionary innovation of protein interactions between core piRNA biogenesis factors and the CtBP (B) and Moonshiner–TFIIA-S–Trf2 (C) transcriptional regulators. The colors of the dotted lines indicate within which species the protein interactions were detected. (D) Proposed model for the functional role of the observed interaction between E(z) and Deadlock from *D. simulans* and *D. erecta*. We speculated that E(z) recruited via Deadlock to Rhino-bound H3K9me3 histones might catalyze local H3K27me3 deposition to reinforce Rhino binding through increased Rhino recruitment and/or residence time (see also "Discussion").

coat color adaptation (Barrett et al, 2019), and toxin resistance evolution in insects (Karageorgi et al, 2019). Furthermore, proteins often function in complex pathways and protein–protein interaction networks, and it remains unclear how adaptive evolution shapes functional pathways composed of multiple connected proteins.

We find that the adaptively evolving genes supporting *Drosophila* germline piRNA biogenesis, in addition to several conserved protein interactions, display at least two layers of innovation of biochemical functionality. First, we showed that while most interactions known from *D. melanogaster* are conserved in the other *Drosophila* species, several show evidence of protein complex interface coevolution (Fig. 6A). Second, we observed rewiring affecting protein interactions in the recruitment of CtBP and Moonshiner–TFIIA-S–Trf2 transcriptional regulators. In this way, the recruitment of these effectors via protein–protein interaction is conserved, but the identity of the connected proteins is evolutionarily fluid.

Of note, there are also rapidly evolving proteins in the network for which we did not uncover innovation in protein–protein interactions (Cutoff and Kipferl). This observation reflects three limitations of the current study: (i) innovation could have occurred for intermolecular interactions beyond the investigated network, (ii) the underlying innovation may concern biochemical properties not tested in the present study, such as protein folding, stability, or fine-tuning of protein–protein interaction affinity, which is not well-captured by the more qualitative nature of Y2H assays (Estojak

et al, 1995), and (iii) it is not possible to directly link the detected positive selection signatures to specific innovations in the interaction network. Future research using unbiased interactome profiling, quantitative affinity measurements, and broad species sampling to increase temporal resolution promises to address these limitations and the connected open questions.

## Species-restricted recruitment of transcriptional regulators to piRNA source loci

Our Y2H data uncovered multiple species-restricted interactions between CtBP and different proteins in the Rhino interaction network (Nxf3, Deadlock, Bootlegger, Kipferl, TFIIA-S, and Moonshiner) (Fig. 6A). We interrogated the in vivo relevance of these interactions by co-IP/MS of *D. simulans* Bootlegger and Deadlock from *D. simulans* and *D. erecta* and revealed strong congruence between the Y2H and co-IP/MS data (Fig. 4). Furthermore, we were able to identify a short linear interaction motif specific to *D. simulans* Bootlegger that is necessary and sufficient for interaction with CtBP (Figs. 5 and EV5). A similar motif has also evolved in *D. erecta* Deadlock (Fig. EV5E), which we found to interact with CtBP in both Y2H and co-IP/MS experiments. Together, our results reveal extensive evolutionary innovation in the recruitment path (i.e., interaction rewiring) of CtBP to Rhino-bound piRNA source loci (Fig. 6B). CtBP recruitment to piRNA source loci has been shown to silence the activity of canonical DNA-encoded promoters, while not affecting

the non-canonical heterochromatin-dependent transcription initiated via Moonshiner (Parhad et al, 2019).

In addition to rewiring of CtBP recruitment, we also identified species-restricted interactions connecting the Moonshiner–TFIIA-S–Trf2 complex to the Rhino interaction network (Fig. 6C). While it remains unclear if the previously proposed recruitment of Moonshiner via Deadlock in *D. melanogaster* (Andersen et al, 2017) is supported by direct interaction between the two proteins, our Y2H as well as co-IP/MS results show a clear interaction between Moonshiner and Deadlock from *D. simulans* and from *D. erecta* (Figs. 4, EV2, and EV4). We propose that (i) Moonshiner complex recruitment via direct interaction with Deadlock is central to the *Drosophila* piRNA pathway and (ii) unknown variation in interaction mode or regulation contributes to the observed differences in Deadlock–Moonshiner interaction across species. Furthermore, in *D. persimilis*, interactions were observed by Y2H between Rhino and both TFIIA-S and Trf2, suggesting that recruitment of the Moonshiner–TFIIA-S–Trf2 complex may be supported by interaction with Rhino in this species. Together with the rewiring of CtBP recruitment, these findings reveal that the interactions mediating recruitment of the key transcriptional regulators of germline piRNA source loci undergo recurrent evolutionary innovation. Interestingly, expression of *D. simulans* Cutoff in *D. melanogaster* was previously found to lead to increased association of Trf2 and CtBP with Rhino-bound loci, suggesting that adaptation of Cutoff also contributes to the recurrent innovation of transcription factor recruitment (Parhad et al, 2019). Given that we did not detect direct interactions between Cutoff and such transcription factors, it seems likely that *D. simulans* Cutoff stimulates transcription factor recruitment indirectly or through complex formation with, e.g., Deadlock. We propose that rewiring of CtBP and Moonshiner–TFIIA-S–Trf2 complex recruitment may reflect adaptive evolution to ensure transposon promoter silencing and content balancing of the piRNA pool during host–transposon arms race evolution (further discussed below).

While our well-controlled Y2H matrix assay by experimental design defines the set of investigated protein–protein interactions, the co-IP/MS experiments also allowed the identification of novel protein interactions. These data uncovered an unexpected enrichment of the E(z) protein in co-immunoprecipitations of Deadlock from *D. simulans*, *D. erecta*, and *D. virilis*, which we did not find with *D. melanogaster* Deadlock (Figs. 4 and Fig. EV4). E(z) is the catalytic subunit of the Polycomb Repressive Complex 2 (PRC2) complex and is responsible for the methylation of K 27 on the N-terminal tail of Histone 3 (Nekrasov et al, 2005; Tie et al, 2007). H3K27me3 modification by PRC2 was recently found to co-occur with H3K9me3 at Rhino-bound piRNA source loci and E(z) depletion was further observed to lead to loss of piRNA production from several Rhino-dependent piRNA source loci (Akkouche et al, 2024). Based on these observations, Akkouche and colleagues have proposed that an H3(K9 + K27)me3 dual-histone mark code acts in parallel with Kipferl to specify Rhino-bound genomic loci. Based on our observed connection between Deadlock and E(z) in several *Drosophila* species, we speculate that E(z) recruitment—potentially outside the context of a PRC2 complex—may function in a Rhino-binding reinforcement loop (Fig. 6D). Specifically, initial recruitment of Rhino may lead to Deadlock-mediated E(z) recruitment and the subsequent establishment of H3K27me3 marks, which then

increases Rhino recruitment or binding stability (Akkouche et al, 2024) to stabilize the genomic region as a piRNA source locus.

## Possible genetic drivers of adaptive evolution and biochemical innovation in the piRNA pathway

Our work uncovers biochemical innovations within an adaptively evolving protein network at an unprecedented scale. What might be the biological advantage of such rapid biochemical innovation in a network of essential genes? At least three broader evolutionary forces may contribute to such rapid evolution despite the essential function of the piRNA pathway. First, the adaptive walk model of evolution (Orr, 1998; Moutinho et al, 2022) states that young genes are expected to evolve faster and with a larger mutational fitness effect because they are further from their fitness optimum than older genes. Adaptive walk effects may, therefore, contribute to the rapid evolution of young, *Drosophila*-restricted piRNA pathway genes such as Rhino, Deadlock, Cutoff, Moonshiner, Bootlegger, and Nxf3. Second, the piRNA autoimmunity model (Blumenstiel et al, 2016; Wang et al, 2020; Kelleher et al, 2020; Kelleher, 2021; Blumenstiel, 2025) proposes that the rapid evolution of piRNA source loci frequently gives rise to piRNAs that target important host genes for silencing. Under this model, such self-targeting would drive adaptation for changes in piRNA pathway proteins that reduce the host gene silencing, e.g., by altering the piRNA biogenesis mechanism or the silencing potency. Third, the evolutionary arms race model suggests that recurrent rapid evolution of piRNA pathway genes occurs in response to transposon evolution toward evasion of piRNA silencing or "anti-silencing" by inhibiting the piRNA pathway. While such "Red Queen evolution" (Van Valen, 1973) has been proposed based on observations of functional diversification of piRNA pathway genes between closely related species (Parhad et al, 2017, 2019; Luo et al, 2020), only the recent identification of a *Drosophila* transposon expressing piRNAs that target a piRNA pathway factor for silencing (Ellison et al, 2020) supports the existence of such piRNA pathway inhibitors. Finally, our data also suggest a fourth, 'piRNA locus balancing', model to explain rapid piRNA pathway evolution. This model rests on the observation that germline piRNA source loci differ in their dependence on specific piRNA precursor biogenesis factors. Some source loci are, for example, expressed independently of Moonshiner and Trf2 and instead rely on DNA-encoded promoters (Andersen et al, 2017). Other loci do not require the zinc finger protein Kipferl (Baumgartner et al, 2022), and a few germline piRNA source loci (e.g., *cluster20A*) are expressed independently of any factors in the Rhino interaction network (Klattenhoff et al, 2009; Mohn et al, 2014; Andersen et al, 2017; ElMaghraby et al, 2019; Baumgartner et al, 2022). We propose altering the interactions between proteins at germline piRNA clusters could impact piRNA production from specific loci. Depending on current transposon activity in a population, altered piRNA sub-pools could provide a mechanism to shift the target focus between transposon families. This model is thus consistent with the recent observation that while piRNA clusters tend to be syntenic between *Drosophila* species, they share little sequence content and instead contain large insertions of recently active transposons (Wierzbicki et al, 2023). Of note, it was recently reported that piRNA pathway genes also show rapid evolution signatures in *D. ananassae* and *D. willistoni*, although to a lesser

extent than D. melanogaster, suggesting that the driver(s) of rapid piRNA pathway evolution are dynamic (Blumenstiel et al, 2025).

In sum, our study on the evolution of protein–protein interactions across *Drosophila* evolution has uncovered principles of interaction evolution within an adaptively evolving protein network. Our findings exemplify how pathway function is maintained despite rapid biochemical innovation and lay the foundation for a better mechanistic understanding of the biochemically labile piRNA pathway and the evolutionary forces driving such innovation. Future studies linking molecular mechanistic and functional characterization to evolutionary analyses will be key to uncovering which evolutionary forces are driving the rapid evolution and biochemical innovation of the piRNA pathway in *Drosophila* and what the functional impact of such innovation is. Amongst such studies, further probing for transposon-encoded silencing evasion or anti-silencing mechanisms, as well as comparative investigations of piRNA source locus regulation by perturbing specific protein–protein interaction in different *Drosophila* species, promise to be revealing.

# Methods

### Reagents and tools table

| Reagent/resource | Reference or source | Identifier or catalog number |
|---|---|---|
| **Experimental models** | | Overview in Appendix Table S1 |
| *Dm*Bootlegger-GFP | ElMaghraby et al, 2019 | |
| GFP in ovaries | ElMaghraby et al, 2019 | |
| GFP-*Dm*Deadlock | This study | FS368 |
| GFP-*Ds*Deadlock | This study | FS369 |
| GFP-*De*Deadlock | This study | FS370 |
| GFP-*Dv*Deadlock | This study | FS371 |
| GFP-*Dm*Bootlegger | This study | FS372 |
| GFP-*Ds*Bootlegger | This study | FS373 |
| GFP-*Dm*Rhino | This study | FS374 |
| GFP-*Dp*Rhino | This study | FS375 |
| *Dm*Kipferl-GFP | This study | FS376 |
| *De*Kipferl-GFP | This study | FS377 |
| **Recombinant DNA** | | |
| pDONR221 | Thermo Fisher | 12536017 |
| pBTM116-D9 | Addgene | 111232 |
| pBTMcC24-DM | Addgene | 111236 |
| pACT4-DM | Addgene | 111233 |
| pCBDU-JW | Addgene | 111237 |
| pAc5.1-OST4-mEGFP | Kubíková et al, 2023 | JK-268 |
| pAc5.1-FspAI-mCherry-Daed-IsoA237-279 | This study | JM-292 |

| Reagent/resource | Reference or source | Identifier or catalog number |
|---|---|---|
| pAc5.1-mCherry | Jeske et al, 2017 | T7-MJ |
| pRASA_NLAP-DmDel | This study | P359 |
| pRASA_NLAP-DsDel | This study | P360 |
| pRASA_NLAP-DeDel | This study | P361 |
| pRASA_NLAP-DvDel | This study | P363 |
| pRASA_NLAP-DmBoot | This study | P364 |
| pRASA_NLAP-DsBoot | This study | P365 |
| pRASA_NLAP-DmRhi | This study | P366 |
| pRASA_NLAP-DpRhi | This study | P367 |
| pRASA_CLAP-DmKip | This study | P370 |
| pRASA_CLAP-DeRhi | This study | P371 |
| **Antibodies** | | |
| Rabbit anti-Rhino polyclonal antibodies | This study | P4 |
| Goat anti-rabbit IgG (H + L) Secondary Antibody, Alexa Fluor 568 | Thermo Fisher | A11011 |
| **Oligonucleotides and other sequence-based reagents** | | |
| PCR primers | This study | Appendix Table S2 |
| **Chemicals, enzymes, and other reagents** | | |
| Gibson Assembly Master Mix | NEB | E2611L |
| iScript gDNA clear cDNA Synthesis kit | BioRad | 1725035 |
| Gateway BP Clonase II Enzyme mix | Thermo Fisher | 11789013 |
| Gateway LR Clonase II Enzyme mix | Thermo Fisher | 11791100 |
| FastDigest Bsp1407I | Thermo Fisher | FD0933 |
| Salmon SS-DNA | Sigma Aldrich | D1626 |
| PefaBloc SC | Sigma Aldrich | 11429868001 |
| IGEPAL CA-630 | Sigma Aldrich | I3021 |
| anti-FLAG M2 magnetic beads | Sigma Aldrich | M8823 |
| DAPI | Biotium | 40009 |
| ProLong Diamond Antifade Mountant | Thermo Fisher | P36961 |
| jetOPTIMUS transfection reagent | Polyplus-transfection | 101000051 |
| µL-Slides 4 Well | ibidi | 80426 |
| **Software** | | |
| DnaSP v6 | http://www.ub.edu/dnasp/; Rozas et al, 2017 | |
| CODEML (PAML) v4 | Yang, 2007 | |
| BLAST | Altschul et al, 1990 | |
| MUSCLE | Edgar, 2004 | |
| Jalview | https://www.jalview.org/ | |
| DIA-NN v1.8.2beta27 | Demichev et al, 2020 | |

| Reagent/resource | Reference or source | Identifier or catalog number |
|---|---|---|
| R | https://www.r-project.org/ (R Core Team, 2023) | |
| VolcaNoseR | Goedhart and Luijsterburg, 2020 | |
| Fiji ImageJ | https://imagej.net/software/fiji/ (Schindelin et al, 2012) | |
| Illustrator | Adobe | |
| Other | | |
| TimsTOF Pro ion mobility mass spectrometer | Bruker | |
| UltiMate 3000 UHPLC system | Thermo | |
| Reversed-phase C18 Aurora column (25 cm × 75 μm) with an integrated Captive Spray Emitter | IonOpticks | |
| LSM980 with Airyscan 2 | Zeiss | |
| AX-CLSM Confocal Microscope System | Nikon | |

## Methods and protocols

### Evolutionary analyses of dual-strand piRNA cluster transcription and export genes

Estimates of dN/dS between *D. melanogaster* and *D. simulans* and those between *D. yakuba* and *D. erecta* were calculated using the software DnaSP (v6 (Rozas et al, 2017)). To test for positive selection, a phylogeny-based test was implemented using the site-specific models in the CODEML program in PAML (v4 (Yang, 2007)). Comparisons of models M7 (neutral: the dN/dS values fit a beta distribution between 0 and 1) and M8 (non-neutral: M7 parameters plus dN/dS >1), as well as a comparison of models M8a (null: the dN/dS values fit a beta distribution plus dN/dS = 1) and M8, were performed. The F3x4 model of codon frequency was used for both comparisons. The degrees of freedom for each test were set to 2 for the M8-M7 comparison and 1 for the M8-M8a comparison, and the log-likelihood ratio test was inferred as significant if the *P* value of the Chi-squared test was less than 0.05. All orthologs were extracted from either FlyBase or NCBI. PAML results and species are listed in Appendix Fig. S1.

### Plasmid cloning by Gibson assembly

Plasmids for ReLo assays and fly injections were assembled, designed in SnapGene (v7.0.3), and cloned via Gibson reactions (NEB; E2611L) according to the instruction manual with 20–25 nt of fragment overhang by default. Reactions were pipetted at room temperature, immediately shifted to 50 °C after adding the reaction mix for 60 min, and afterward transformed into DH5α chemically competent bacteria. In total, 50 μL bacteria were thawed on ice, and 2 μL of the pre-chilled Gibson reaction was used to transform bacteria. The mixture was gently flicked and then incubated for 20' at room temperature, transferred to 42 °C for 42 s, and then chilled on ice for 2 min. 200 μL of standard LB medium (10 g/L NaCl, 10 g/L peptone from casein, 5 g/L yeast extract) was added, and the cells were incubated at 37 °C at 550 rpm for 1 h. Cells were then plated on antibiotic-containing LB-agar (LB medium with 15 g/L agar) plates and incubated overnight at 37 °C.

### Ortholog identification

Orthologous genes to the *D. melanogaster* piRNA pathway genes were identified as reciprocal best hits from BLAST (Altschul et al, 1990) searches using the NCBI reference genome for each species, followed by synteny analysis (Appendix Figs. S2–5). For synteny analysis, we identified the genomic coordinates of each identified ortholog in HiC-scaffolded chromosome-level genome assemblies (described in (Gebert et al, 2025)) by BLAST. We then used gene annotations based on orthology to D. melanogaster genes to plot gene synteny in 100 kb windows around each gene in the five investigated genomes. Of note, we did not find evidence of synteny at the reciprocal best BLAST hits for *D. melanogaster* Moon (DmMoon) and DmNxf3 in *D. persimilis*, indicating potential relocation by recombination or duplication and subsequent loss of the original copy. For the protein sequence alignment for CtBP (Appendix Fig. S8), sequences were retrieved from NCBI using the noted accession numbers, aligned using the MUSCLE program (Edgar, 2004), and displayed using Jalview.

### Plasmid cloning for yeast-two-hybrid assays

Coding sequences of the orthologs were pulled from NCBI and synthesized by Twist Bioscience, except all Deadlock orthologs, which were amplified from cDNA of *Drosophila* ovaries. One μg of total ovary RNA was DNase-treated for cDNA amplification and reverse transcribed using the iScript gDNA clear cDNA Synthesis kit (BioRad, 1725035). Coding sequences were then amplified from cDNA using sequence-specific primers containing overhangs for Gibson assembly (see Appendix Table S2 for PCR primer sequences). Correctly sized amplicons were used for Gibson reactions to clone them into Gateway vectors.

The 53 coding sequences of five different *Drosophila* species were cloned into pDONR221 (gift of Will Garland, Torben Heick Jensen lab, Aarhus University) with BP clonase II (Thermo Fisher, #10348102). Each CDS was then cloned into two Gateway compatible bait (Addgene #111232 (pBTM116-D9) and #111236 (pBTMcC24-DM)) & two prey (Addgene #111233 (pACT4-DM) and #111237 (pCBDU-JW)) vectors (Woodsmith et al, 2017) with LR clonase II (Thermo Fisher; 11791100). LR reactions were performed according to standard protocols and two clones of each construct were picked for DNA mini-preparation (Hegele et al, 2012; Worseck et al, 2012). Each plasmid was digested with Bsp1407l to assess the expected insert size by agarose electrophoresis.

### Yeast growth and transformation

Yeast strains were grown on either yeast-extract-peptone-dextrose-adenine (YPDA) medium (content (w/v): 1% bacto-yeast extract; 2% bacto-peptone; 2% glucose; 0.004% Adenine) or supplemented nitrogen base (NB) (1.33% yeast nitrogen base (w/v)) for growth under selection. Supplemented NB was in each case added 2% Glucose, 0.004% Adenine, 0.002% Uracil, and the following specific to the use case: (i) Growth of bait vectors: 0.002% each of Leucine, Histidine, and Methionine. (ii) Growth of prey vectors: 0.002%

each of Tryptophane, Histidine, and Methionine. (iii) Mating Control: 0.002% each of Histidine and Methionine. (iv) Auto activity test of baits: 0.002% each of Leucine and Methionine.

A correctly sized replicate plasmid of the shuffled plasmids was used to transform Mat-a (bait vectors, Trp selection) or Mat-α (prey vectors, Leu selection). The bait strain genotype was *his3Δ300; trp1-901; leu2-3,112; ade2; GAL4; can1; cyh2; URA3::(lexAop)8-GAL1TATA-lacZ; LYS2::(lexAop)4-HIS3TATA-HIS3*. The prey strain genotype was *his3Δ300; trp1-901; leu2-3, 112; ade2; lys2-801am; gal4; gal80; cyh2; can1; ura3::(lexAop)8-GAL1TATA-lacZ; LYS2::(lexAop)4-HIS3TATA-HIS3; ADE2::(lexAop)8-GAL1TATA-URA3*.

Pre-transformation yeast cultures were grown overnight at 30 °C in 10 mL YPDA medium while shaking at 130 rpm. On the following day, the ODs of yeast strains were measured, and 60 mL (60 mL are sufficient for one 96-well plate) main cultures were infected to OD 0.13 and grown as before. At OD 0.5–0.8, cells are harvested by centrifuging for 5 min at $1000 \times g$ and washed once in PBS. In the meantime, flat bottom 96-well plates were prepared with ~350 ng plasmid DNA and 25 µg salmon SS-DNA (Sigma Aldrich, D1626). Yeast cells were resuspended in 2.2 mL Mix 1 (0.1 M Lithium Acetate; 5 mM Tris-HCl pH 8.0, 0.5 mM EDTA; 1 M Sorbitol) and incubated for 20 min at room temperature. Yeasts were resuspended again, and 22 µL of the mix was added to each well, prepared with DNA. The mix was mixed by vortexing the plate. Next, 120 µL of Mix 2 (0.1 M Lithium Acetate; 10 mM Tris-HCl pH 8.0; 40% PEG3350) were added, wells were mixed, and the plates were incubated at 30 °C for 30'. Then 16 µL DMSO was added, wells were mixed by vortexing, and plates were heat-shocked at 42 °C for 15 min. Finally, each transformant was stamped out four times. Four biological replicates of each transformed strain were maintained throughout the screen.

### Yeast-two-hybrid matrix screening

The basic workflow of the screen was performed as described (Hegele et al, 2012; Worseck et al, 2012). All tables, including raw counts and processing, are available as Microsoft Access or Excel files on request. A custom gridding robot (kbiosystems) was used to stamp yeast onto agar growth plates in 96- and 384-well formats. Mating of haploid yeasts was performed on YPDA medium. A nitrogen base medium, supplemented with amino acids and nucleotides, was used for selective growth. All solid trays and plates for yeast growth were supplemented with 2% Agar to solidify. Individual media are listed above under "Yeast growth and transformation".

Auto-active bait constructs were identified by stamping each transformed strain onto -His trays without previous mating. Auto-active bait constructs were excluded from the analyses. A 384-format prey strain matrix (Fig. EV1) was assembled from 3 to 4 replicas of prey constructs stamped out on -Leu trays and grown for three to four days before mating. Bait strains were grown in liquid culture (15–20 h, 30 °C, 150 rpm). The prey matrix was transferred into the liquid bait culture, mixed, and transferred to YPDA for mating. Mating was performed on medium for 20–28 h. Colonies were then transferred directly to -His/-Trp/-Leu/-Ura/-Ade medium for protein interaction selection. Colonies were incubated over seven days, and images were taken daily from day four onward. Colonies were counted with a growth score of 1 or 3, 1 typically being a single small colony (background) and 3 a fully grown yeast spot. Furthermore, we excluded two prey strains from the analyses,

which grew repeatedly with many independently mated bait constructs (prey auto activity). We then counted the colonies with a growth score of 3, combined with technical replicates. We built a ratio with the number of tests performed with the given Y2H construct combination. We got a score with a maximum of 1 for each vector combination in each biological replicate. If all biological replicates showed the interaction, we went on with the highest score found in a specific biological replicate. Lastly, we collapsed all vector combinations into protein–protein interactions and added the individual scores of eight possible vector combinations to achieve a maximal score of eight. This score of eight represents a protein–protein interaction detected in all eight vector combinations and technical replicates of at least one biological replicate.

### Fly husbandry

Fly stocks were maintained at 25 °C with 70% humidity in 12:12 h light/darkness cycles. For maintenance and experiments, flies were fed on standard fly medium following the VDRC "Normal" fly food media recipe. All fly lines used in this study are listed in Appendix Table S1. Of note: *D. persimilis* Deadlock is missing from the complete set of Deadlock genes because embryo injections failed.

### D. melanogaster fly lines for transgene expression

PhiC31 integrase-mediated transgenesis via microinjection was performed by The University of Cambridge Department of Genetics Fly Facility. Briefly, plasmids (Appendix Table S1) contained a 70 bp *attB* site for site-specific recombination with attP landing sites and the mini-white ($w^+$) eye-marker gene for selection. Constructs were integrated on chr2 (attP40; FlyBase ID: FBti0114379) and balanced over *CyO* with by two crosses to if/CyO females to remove the chrX-linked PhiC31 integrase transgene. The exogenous coding sequence is expressed under the control of the *D. melanogaster rhi* promoter, a partial *nos* 5'-UTR, and the *vasa* 3'-UTR. The construct is flanked by *gypsy* insulators to establish chromatin borders. Coding sequences were tagged with a localization and purification (LAP)-tag consisting of 3xFLAG, 3xV5, and eGFP for either N-terminal (NLAP) or C-terminal (CLAP) tagging. The tagging terminus was chosen based on the Y2H results using N- and C-terminally tagged bait and prey. Rhi, Del, and Boot proteins were tagged N-terminally, and Kipf proteins were tagged at the C-terminus.

### Co-IPs from FLAG-tagged proteins for mass spectrometry

Fly lines FS21, FS52, FS368, FS369, FS370, FS371, FS372 and FS373 (Appendix Table S1) were used for Co-IP MS sample preparation. Each fly line harbors a transgene expressing the protein of interest fused to either 3xFLAG-3xV5-eGFP- (NLAP) or -eGFP-3xV5-3xFLAG (CLAP). The tagged transgenic proteins were immuno-purified by targeting the 3xFLAG epitopes. After hatching, flies (1 to 2 days old) were placed in cages on apple juice plates (25% (v/v) apple juice; 15 g/l agar; 25 g/l sucrose; 0.15% Nipagin) with yeast paste (dry yeast stirred in tap water until it becomes a sticky paste). Apple juice plates were exchanged twice daily for efficient egg laying and oogenesis. When flies were 3–5 days old, ovaries were dissected. All ovaries were placed immediately on ice in Ephuzzi Beadle Ringer solution (EBR; 10 mM HEPES pH 7.3; 130 mM NaCl; 5 mM KCl; 2 mM CaCl$_2$). Isolated ovaries were collected for a maximum of 1 h before discarding the EBR solution and snap-freezing in liquid nitrogen. Co-immunoprecipitations of all samples

were performed in biological quadruplicates of approximately 200 ovary pairs per replicate. The sample number of four was chosen based on the variance observed in previous Co-IP MS analyses. All buffers and utensils were pre-chilled on ice before usage. The whole following workflow was performed at 4 °C in a cold room. After thawing ovaries on ice, lysis was achieved by douncing (~20 times) in 1 mL ovary protein lysis buffer 2 (OPLB2; 20 mM Tris-HCl pH 7.5, 150 mM NaCl, 2 mM MgCl$_2$, 10% (v/v) Glycerol, 1 mM dithiothreitol* (DTT), 1 mM PefaBloc SC* (Sigma Aldrich, 11429868001), 0.2% (v/v) IGEPAL CA-630 (Sigma Aldrich, I3021) on ice until the solution is 'smooth' and seems homogenous. Reagents marked with '*' were added immediately before usage. The lysates were centrifuged at 21,000 × g for 5' at 4 °C, the fatty layer on top was carefully removed, and clear supernatants were transferred to fresh low-retention tubes (Fisher Scientific, 11986955). Next, anti-FLAG M2 magnetic beads (Sigma Aldrich, M8823) were equilibrated in OPLB2 by placing them on a magnet, replacing the bead storage buffer (20 mM Tris-HCl pH 7.5, 150 mM NaCl) and washing them twice in OPLB2. Finally, beads were resuspended in 4:1 lysis buffer:storage volume. 16 μL of OPLB2 equilibrated beads (4 μL beads in storage volume per sample) were added to each sample. Protein capture was performed at 4 °C while rotating for 3 h. Next, samples were washed four times with 1 mL of OPLB2. Beads were pelleted on a magnet, the supernatant was taken off, the beads were resuspended in 1 mL fresh OPLB2, and samples were rotated for 5 min each. Afterward, samples were additionally rinsed seven times with ovary lysis wash buffer (20 mM HEPES pH 7.4, 2 mM MgCl$_2$, 150 mM NaCl) to get rid of detergents, glycerol, and other molecules interfering with MS. For the first wash beads were resuspended fully and the tubes inverted. In the rinse buffer, beads accumulated at the magnet for 5 min. During the next washes, the beads were kept on the magnet, the tubes were closed, and the rack was inverted. During wash number four, the beads were resuspended fully and transferred to fresh low-binding tubes. After finishing all wash and rinse steps, the supernatant was removed thoroughly. To elute bound proteins, the beads were resuspended in 50 μL 2% sodium dodecyl sulfate by pipetting and rotated for 15 min at room temperature. The eluate was collected, and the elution step was repeated. Both eluates were combined to yield 100 μL Co-IP protein samples. The eluates and the beads were snap-frozen in liquid nitrogen and stored at −70 °C until they were shipped on dry ice.

## MS data acquisition and processing

IP samples (biological quadruplicates) were alkylated with 5 mM TCEP, reduced with 10 mM CAA, and digested with trypsin (~1:10 (wt/wt)). Samples were purified utilizing the S-Trap micro columns (Protifi), following the manufacturer's instructions: S-Trap™ micro spin column digestion protocol. Peptides were lyophilized, resuspended in 0.1% formic acid, and diluted to a concentration of 200 ng/μL, whereby 1 μL was used per MS-injection. Samples were analyzed on a TimsTOF Pro ion mobility mass spectrometer (Bruker) in line with an UltiMate 3000 UHPLC system (Thermo). Peptides were separated on a reversed-phase C18 Aurora column (25 cm × 75 μm) with an integrated Captive Spray Emitter (IonOpticks). Mobile phases A 0.1 vol% formic acid in water and B 0.1 vol% formic acid in ACN (Fisher Scientific, 10799704) with a 300 nl/min flow rate, respectively. Fraction B was linearly increased from 2 to 25% in a 90-min gradient, increased to 40% for 10 min, and further increased to 80% for 10 min,

followed by re-equilibration. The spectra were recorded in diaPASEF mode (Meier et al, 2020). Samples were recorded in technical replicates.

Subsequently, the in DIA mode recorded spectra were quantified with DIA-NN v1.8.2beta27 (Demichev et al, 2020) using a synthetic library built from reviewed and selected unreviewed protein sequences from the UniProt *Drosophila melanogaster* (fasta download: 04.03.2024). The protein database included transgenes and unreviewed proteins of the piRNA core complex. Expression values were log$_2$-transformed, normalized across all protein groups of all samples, and averaged across their technical replicates. Further analysis only considered protein groups identified in both technical replicates. eGFP ratios for proteins of interest were calculated by normalization to the average relative GFP abundance within each sample group. For box plots of relative protein enrichment, whiskers denote 1.5 times the interquartile range (IQR) beyond the first and third quartiles, with any points beyond this range considered outliers (Tukey's Fences/IQR method). Data were visualized using R (R Core Team, 2023), associated packages ggplot2, and VolcaNoseR (Goedhart and Luijsterburg, 2020).

Briefly, in the proteomics dataset, DIA acquisition revealed quantitative data for 523 distinct protein groups, 517 of which were detected in both technical replicates. Approximately half of the detected protein groups (292 out of 517) were detected in every sample group, and 166 protein groups were present in each sample. Conversely, 44% of the protein group abundance values (225 out of 517) displayed varying occurrences across the indicated sample groups.

## Fixed immunofluorescence of ovaries

One to two days old flies (5 males and 20 females) were sorted and transferred to cages on apple juice plates with yeast paste. Flies were cultivated like this at 25 °C for two to three days. Apple juice plates were exchanged twice daily to allow efficient egg laying. Ovaries of three to 5-day-old female flies were then dissected into 200 μL EBR on ice. After a maximum of 30 min on ice, ovaries were fixed for 15 min at room temperature while rotating after adding 1 volume (200 μL) of 4% paraformaldehyde (final concentration 2%). Next, ovaries were washed 3 times for 5 min at room temperature each in 750 μL PBX under rotation (PBX: PBS with 0.3% Triton X-100). During the second wash, ovaries were dissociated into ovarioles by pipetting ~10 times with a 200 μL tip. After washing, ovarioles were blocked in 750 μL BBX (1% (w/v) Bovine Serum Albumin in PBX) for 2 h at room temperature while rotating. For endogenous Rhino staining using fluorescent antibodies, the samples were covered in aluminum foil. Ovarioles were rotating in BBX with primary rabbit anti-Rhino (raised against the SRNHQRPNLGLVDAPPNDHVEE peptide, similar to (Mohn et al, 2014) immunization and purification done by Davids Biotechnologie GmbH, Germany) antibodies at 4 °C overnight. The next day, ovarioles were washed four times with 750 μL PBX at room temperature while rotating. Next, ovarioles were incubated in secondary antibody (Goat anti-Rabbit IgG (H + L) Secondary Antibody, Alexa Fluor 568, Thermo Fisher, A11011) for 2 h and washed four times as before. DAPI (Biotium, 40009) was added to a final concentration of 0.5 μg/ml during the second wash step. Lastly, ovarioles were mounted on slides with ProLong Diamond Antifade Mountant (Thermo Fisher, P36961), covered with coverslips, and sealed with nail polish. The mounting medium was allowed to harden for 24 h at room

temperature in the dark before microscopy. For long-term storage, slides were kept at 4 °C in the dark. Images were acquired with a Zeiss LSM980 in Airyscan 2 (General settings: Lasers: 488 nm 5.0% intensity for Deadlocks and 3.5% intensity for Bootleggers, 850 V master gain; 543 nm 2.0% intensity, 810 V master gain; 405 nm 0.5% intensity, 808 V master gain). All images were processed by the software's Airyscan2 processing in batch mode with default settings ('Standard' strength). Acquired images were adjusted (Contrast, Brightness, Maximum, Minimum) in Fiji ImageJ (Schindelin et al, 2012).

### ReLo protein–protein interaction assays

The experiments were performed as described in (Salgania et al, 2024a). Briefly, we used pAc5.1-OST4-mEGFP-FspAI (pAc5.1-OST4-mEGFP; JK-268; (Kubíková et al, 2023)) as the basis for the bait vectors and pAc5.1-FspAI-mCherry-Daed-IsoA237-279 (pAc5.1-mCherry-MLS; JM-292) as the basis for prey vectors. OST4 is described to anchor tagged proteins to the ER (Möckli et al, 2007). The piRNA processing factor Daedalus (Daed) is anchored to the outer mitochondrial membrane (Munafò et al, 2019) via a transmembrane domain (Daed$^{TMM}$) comprising amino acid residues 237-279 (Salgania et al, 2024b), which was therefore used as an outer mitochondrial membrane anchor. The pAc5.1-mCherry-MLS plasmid was generated by inserting the Daed MLS (aa 237-279) into the EcoRV site of the pAc5.1-mCherry vector (T7-MJ; (Jeske et al, 2017)) with subsequent insertion of a unique FspAI blunt-end restriction site 5' of the mCherry coding sequence. Backbones were cut using FspAI, and inserts were ligated using Gibson reactions. For prey constructs, 5'-agagaccccggatcgggtGCCAAC-3' was used as an overhang on the forward primer (capital letters are the added Kozak sequence), and 5'-ttggtaccgccgctggatgc-3' was used as an overhang on the reverse primer. For bait constructs, 5'-gcttgaattctgcaacgtgc-3' was used as an overhang for the forward primer, and 5'-gcatgagatatccagcacag-3' was used as an overhang for the reverse primer. S2R+ cells were split to a $10^6$ cells/mL density two days before transfection. Cells were rinsed off, resuspended, and counted. In total, 600,000 S2R+ cells were seeded on µL-Slides 4 Well IbiTreat slides (Ramcon; 80426) in 600 µL medium and co-transfected 2 h after seeding with 300 ng per plasmid (600 ng total) by using the jetOPTIMUS transfection kit (VWR; 101000051) according to the instruction manual. Cells were imaged 30 to 48 h post-transfection without fixation with a Nikon AX-CLSM (Heidelberg, Germany) or Zeiss Axio LSM980 (Aarhus, Denmark) in confocal imaging mode. For quantification, co-localization patterns were categorized blinded on randomized images.

## Data availability

The datasets produced in this study are available in the following databases: Protein interaction AP-MS data: PRIDE PXD053049. An author checklist adapted from the MDAR (Materials Design Analysis Reporting) Framework for transparent reporting in the life sciences has been followed and is provided along with the paper.

The source data of this paper are collected in the following database record: biostudies:S-SCDT-10_1038-S44318-025-00439-8.

## Peer review information

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

## Acknowledgements

We appreciate the helpful comments on earlier manuscript versions from Julius Brennecke and Anne F. Nielsen. We thank the Nikon Imaging Center of the University of Heidelberg for access to microscopes. We thank Jutta Metz for generating the pAc5.1-mCherry-MLS vector. We thank Teixeira FK, Ravindra-Kumar T, and Gebert D for generously sharing data for synteny analyses before publication. SR was supported by an EMBO scientific exchange grant (#10132) and the Aarhus University GSNS Mobility Grant. Work in the Jeske lab was funded by the Emmy Noether Program of the German Research Foundation (DFG; JE-827/1-1 to MJ). The Andersen group is supported by the Novo Nordisk Foundation (NNF18OC0030954), the Independent Research Foundation Denmark (9064-00056B), and an AIAS Marie Curie CO-FUND Fellowship (PA). The work was also supported by BioTechMed-Graz, project DYNIMO, and the Austrian Science Fund, FWF (https://doi.org/10.55776/P30162, https://doi.org/10.55776/P34316), as well as the Field of Excellence BioHealth—University of Graz.

## Author contributions

**Sebastian Riedelbauch**: Conceptualization; Data curation; Formal analysis; Validation; Investigation; Visualization; Methodology; Writing—original draft; Project administration. **Sarah Masser**: Data curation; Formal analysis; Investigation; Visualization; Methodology; Writing—review and editing. **Sandra Fasching**: Investigation; Methodology. **Sung-Ya Lin**: Data curation; Formal analysis; Writing—review and editing. **Harpreet Kaur Salgania**: Investigation; Methodology. **Mie Aarup**: Investigation. **Anja Ebert**: Data curation; Validation; Investigation; Visualization; Writing—review and editing. **Mandy Jeske**: Supervision; Methodology; Writing—review and editing. **Mia T Levine**: Data curation; Formal analysis; Supervision; Writing—review and editing. **Ulrich Stelzl**: Formal analysis; Supervision; Visualization; Methodology; Writing—review and editing. **Peter Andersen**: Conceptualization; Formal analysis; Supervision; Funding acquisition; Visualization; Writing—original draft; Project administration; Writing—review and editing.

Source data underlying figure panels in this paper may have individual authorship assigned. Where available, figure panel/source data authorship is listed in the following database record: biostudies:S-SCDT-10_1038-S44318-025-00439-8.

## Disclosure and competing interests statement

The authors declare no competing interests.

# Expanded View Figures

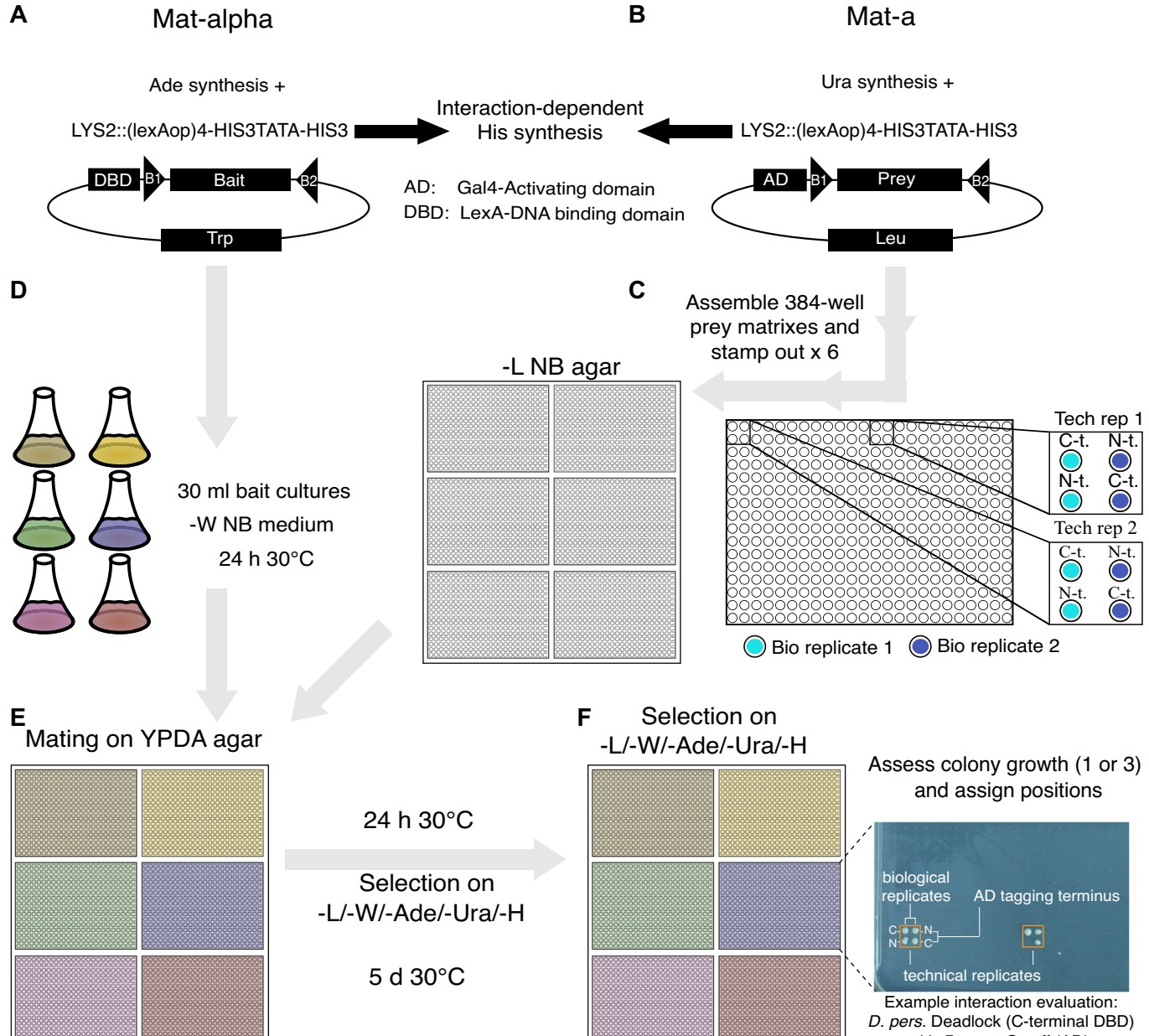

**Figure EV1. Yeast-two-hybrid screen workflow.**

(**A**, **B**) Mat-alpha and Mat-a strains of *S. cerevisiae* were transformed with bait (**A**) and prey (**B**) vectors, respectively, and grown in a selective medium. (**C**) Prey yeast strains were assembled into a fixed 384-well format stamped out in six replicates per tray and grown on selective medium. For mating, the prey matrices were stamped into liquid cultures of bait strains (**D**) and the mixture was transferred to non-selective YPDA medium (**E**). (**F**) After growth, the mated colonies (one bait vector per 384-format prey matrix) were stamped onto limited medium (-Leu: prey plasmid; -Trp: bait plasmid; -Ade: selection against Mat-alpha; -Ura: selection against Mat-a; -His: selection for PPI). The trays were then incubated at 30 °C for four – seven days before images were taken.

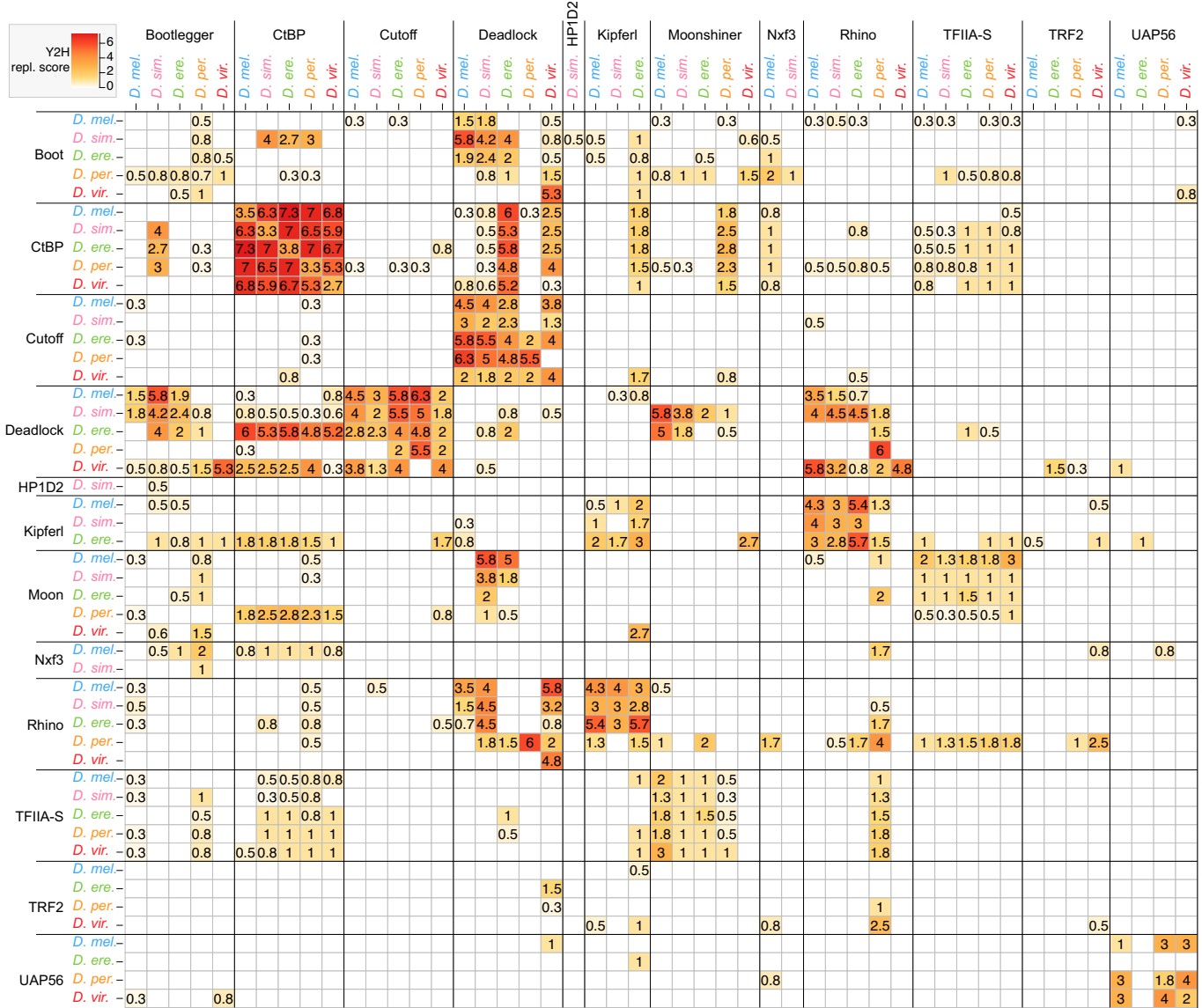

**Figure EV2. Yeast-two-hybrid replication scores for all tested interactions.**

Y2H replication scores are noted in the matrix as numbers and by heat map color code (see legend). The replication scores were calculated as described in Appendix Fig. S1. Of note, HP1D2 was included because it is a likely duplication of Rhino present in *D. simulans*. Since no interactions were found connecting it to the piRNA pathway, we left it out of the main text but included it here for completeness.

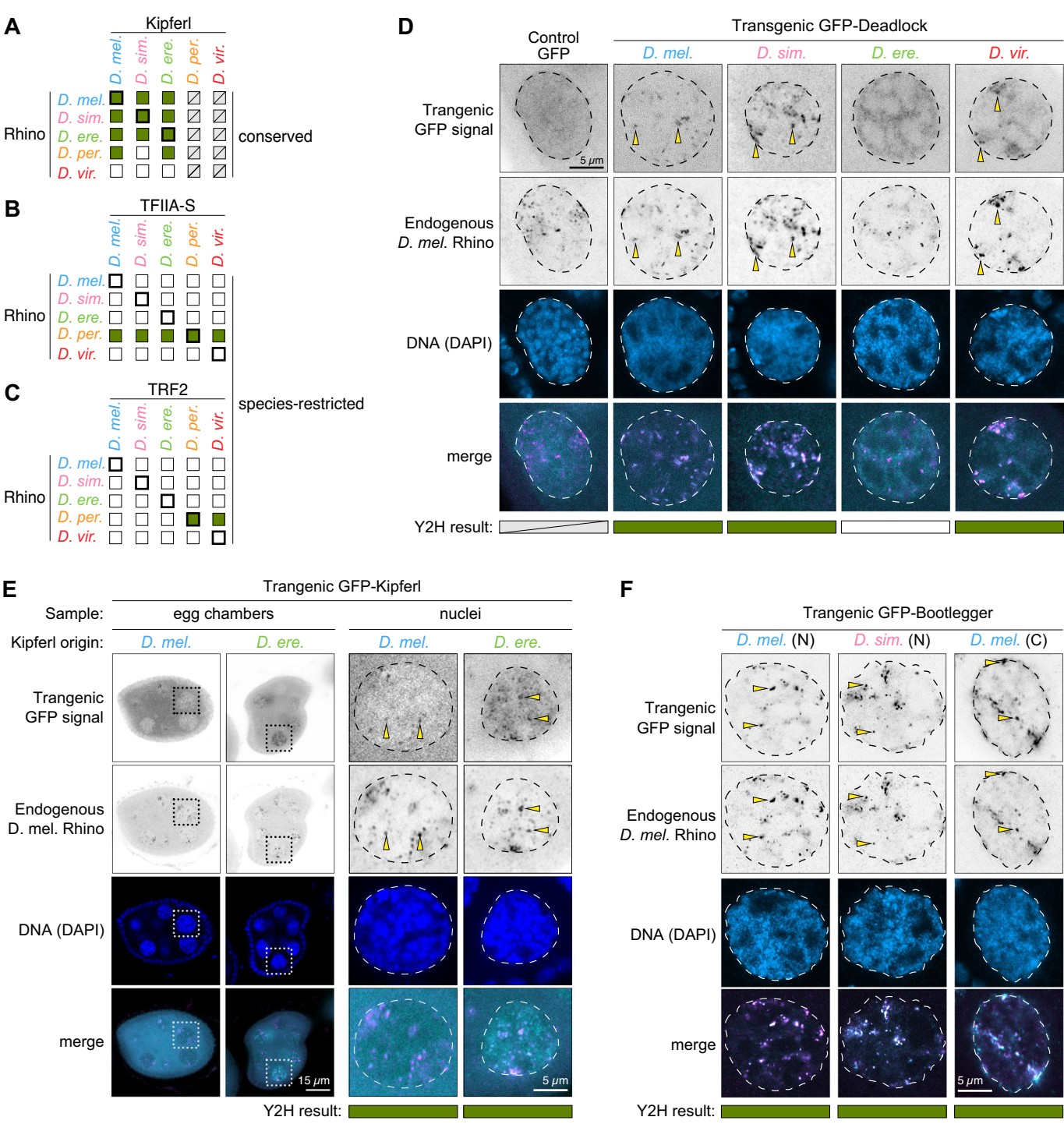

Figure EV3. **Supplementary data related to Fig. 3.**

(A–C) Summary of interactions detected by yeast-two-hybrid between orthologs of rhino and kipferl (A), TfIIA-S (B), and Trf2 (C). Intra-species (thick box outline) and inter-species (thin box outline) interactions are shown as filled green boxes. In contrast, empty boxes indicate the absence of interaction detection above the replication score threshold. Gray crossed-out circles for Kipferl denote the absence of the gene in those species. Pattern: evolutionary signature of the interaction (see main text for details). (D–F) Confocal microscopy images showing the localization of endogenous *D. melanogaster* Rhino (anti-Rhino IF) and GFP-tagged transgenic Deadlock (D; gray-scale images reused from Fig. 3H), Kipferl (E; gray-scale nuclei images reused from Fig. 3I), and Bootlegger (F) from the indicated species. Dashed line: nuclear border as determined by DAPI staining. Yellow arrows highlight co-localizing foci of endogenous Rhino IF signal and transgenic GFP-tagged proteins. Scale bars indicate 5 μm. Source data are available online for this figure.

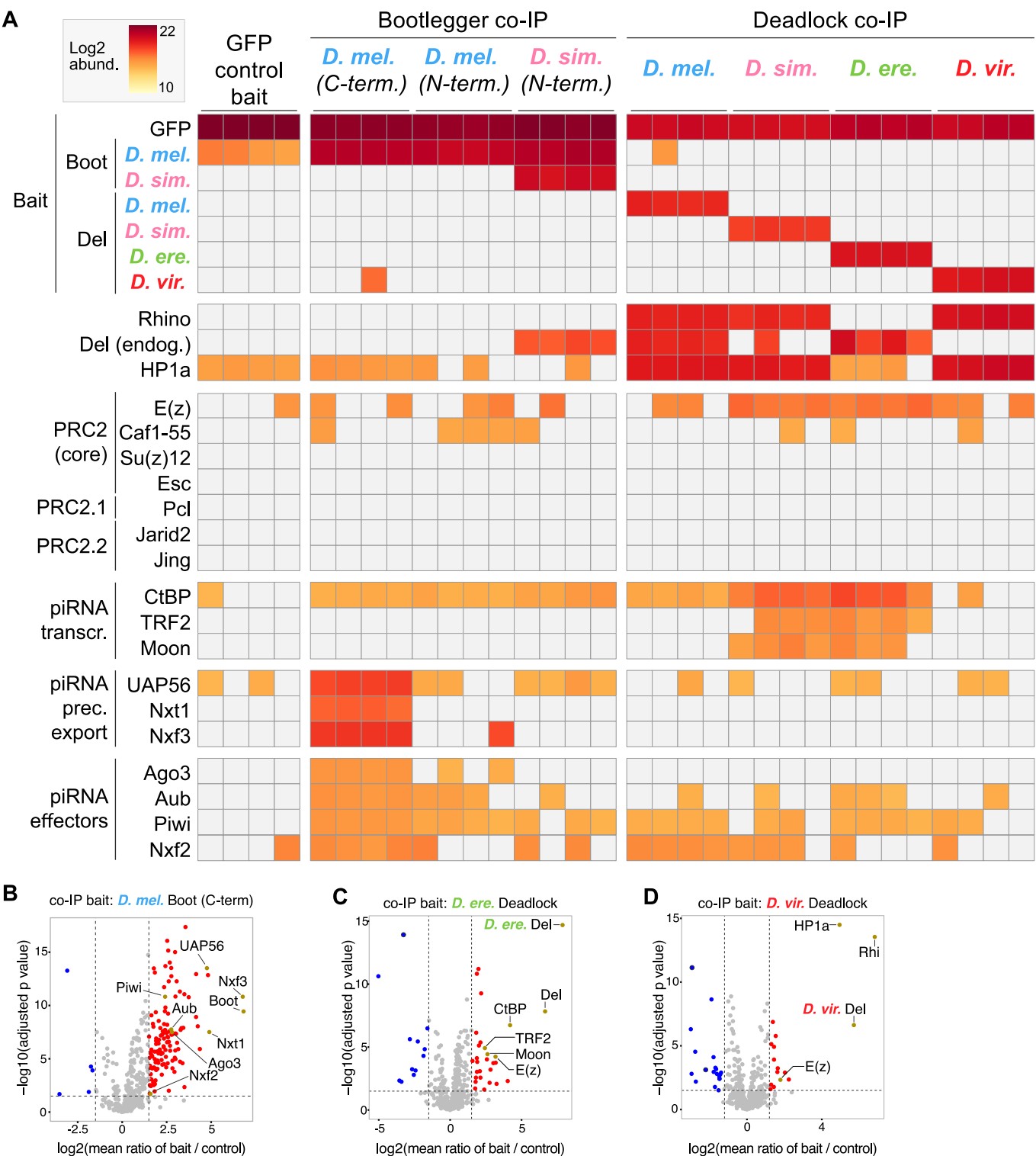

**Figure EV4. Supplementary co-IP/MS data related to Fig. 4.**

(A) Heatmap diagram showing log2-transformed mass spectrometry abundance signal values for the proteins indicated to the left in the co-IP/MS bait samples denoted above the diagram. (B–D) Volcano plots displaying co-IP/mass spectrometry data from the co-IP baits indicated above each plot. The x axes show log2 fold change between bait and control IP averaged over four biological replicates with each two technical replicates. The y axes display the negative log10 of adjusted P values from t tests for enrichment in bait compared to control co-IP. Red and blue dots represent proteins with more than 1.5-fold change with -log10(adjusted P values) higher than 2. Source data are available online for this figure.

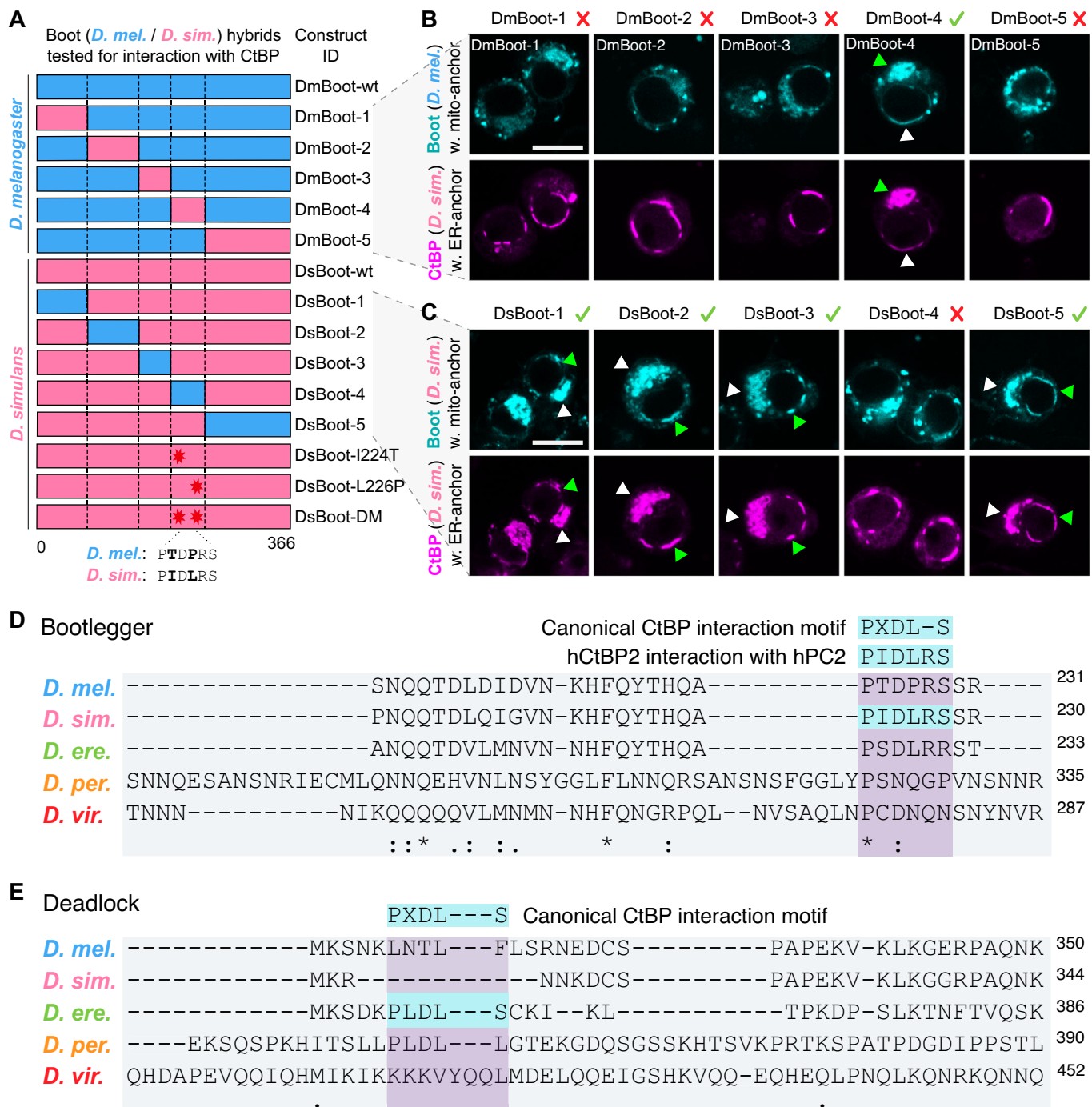

**Figure EV5.   Supplementary ReLo protein interaction analyses related to Fig. 5.**

(A) Schematic representation of domain-swap constructs between the *D. melanogaster* (blue) and *D. simulans* (pink) Bootlegger orthologs with the interaction results indicated to the right by red crosses or green check marks. Black stars indicate the tested interaction site point mutants. (B, C) Fluorescence microscopy images of ReLo protein interaction assays testing interaction between CtBP and *D. melanogaster* Bootlegger proteins harboring swapped domains from *D. simulans* (B) or vice versa (C); see also (A). Shown are representative cells from >10 images. Green arrows indicate protein accumulation in interrupted ring structures around the nucleus. White arrows show protein accumulation at cytoplasmic sites. Scale bars indicate 8 µm in size. Red crosses and green check marks denote whether protein interaction was concluded or not, respectively, based on the ReLo assays. (D, E) Amino acid sequence alignment around the identified CtBP interaction motifs in Bootlegger (D) and Deadlock (E) from the five investigated *Drosophila* species. Source data are available online for this figure.

