## [Peer Review File · The EMBO Journal]

Recurrent innovation of protein-protein interactions in the *Drosophila* piRNA pathway

Sebastian Riedelbauch, Sarah Masser, Sandra Fasching, Sung-Ya Lin, Harpreet Kaur Salgania, Mie Aarup, Anja Ebert, Mandy Jeske, Mia Levine, Ulrich Stelzl, and Peter Andersen

Corresponding author: Peter Andersen (pra@mbg.au.dk)

Review Timeline:

Submission Date:	22nd Oct 24
Editorial Decision:	16th Nov 24
Revision Received:	21st Feb 25
Editorial Decision:	15th Mar 25
Revision Received:	22nd Mar 25
Accepted:	3rd Apr 25

Editor: Yehu Moran

Transaction Report:

Dear Dr. Andersen,

Thank you for submitting your manuscript for consideration by the EMBO Journal. It has now been seen by three referees whose comments are shown below.

Given the referees' positive recommendations, I would like to invite you to submit a revised version of the manuscript, addressing the comments of all three reviewers. I should add that it is EMBO Journal policy to allow only a single round of revision, and acceptance of your manuscript will therefore depend on the completeness of your responses in this revised version.

I would strongly suggest that after going carefully over the referees' comments and discuss them with your co-authors you will prepare a revision plan and send it to me via email in the next few weeks. From our experience this helps in preparing a feasible revision plan and setting realistic expectations between authors and editors and hence significantly improves the chances for a smooth and positive revision process as well as eventual manuscript acceptance.

Thank you for the opportunity to consider your work for publication. I look forward to your revision.

Yours sincerely,

Yehu Moran
Academic Editor
The EMBO Journal

We realize that it is difficult to revise to a specific deadline. In the interest of protecting the conceptual advance provided by the work, we recommend a revision within 3 months (14th Feb 2025). Please discuss the revision progress ahead of this time with the editor if you require more time to complete the revisions.

Referee #1:

This manuscript presents an analysis of protein-protein interaction among components of the piRNA biogenesis pathway across 5 species of *Drosophila*. The authors identify cases of conserved interaction (proteins interact within and among species), coevolved interaction (proteins interact within but not among species), and changes in pathway 'wiring' (interactions vary among species). They discuss these findings in the light of ongoing hypotheses about the, unusually fast, evolution of piRNA pathway components in *Drosophila*.

This paper represents an large and impressive piece of work. The logic seems sound, and the findings will be of great interest to anyone working on piRNAs or pathway evolution more generally. The work is well motivated, and the manuscript written and presented very well.

Any major concerns will, I think, arise from the reliability of Y2H screening - which I am not well placed to assess critically. However, it does appear that they have taken as robust an approach as possible, and because many of the claims are further supported using CoIP/Mass Spec. and/or co-localisation based on microscopy, my own feeling is that the overall take-home messages are robust and interesting, even if some interactions (and especially missing interactions) may prove to be spurious.

Below, I list some individual points below (most trivial), in line order

L38 adaptive evolution is a response to positive selection, and therefore a marker of it - but the terms are not synonymous. L48 emphasise that few /experimental/ studies have done so

L93, L100, 159 and elsewhere - many of the in-text citations are mis-formatted with author initials. This is a bug in Zotero.

Introduction: Overall is a bit wordy - I think it could be shortened by 20% with no loss of clarity.

L128-153 and 560-568. This section on evolutionary rates is a bit weak. Most notably, (i) Ka/Ks between Dper and Dpse will be very noisy and dominated by polymorphism, as Ks is only ~3%. Such an analysis should aim for species between Ks=10% and 30% divergent. (ii) The models are insufficiently described in the results, and in any case M7 vs M8 is not good, and best practice would be M8A vs. M8. (iii) I am very concerned that, because these proteins are so divergent, the inference from PAML models will be spurious due to misalignment (AS noted in Figure S1 legend). How good do the alignments look in regions that have inferred selection? Best Practice might be to use BAliPhy (<https://doi.org/10.1093/bioinformatics/btab129> and <https://doi.org/10.1093/molbev/msu174>)

L193-197 I am not equipped to assess whether the 'multimeric complexes' argument is acceptable, or a case of special pleading.

L233-239. I am dubious about the robustness of the claim "... at a similar high frequency ..." In general, I would not accept such a claim without an explicit statistical test. However, after several minutes thought I cannot formulate one, so perhaps the authors could not either.

L250-252 "In sum ... are characterized by either conserved, co-evolving, or species-restricted interactions" - does this mean "anything can happen" ?

L286-296 I found this section hard to follow, as the species of origin was not always explicit in each case that a Dmel gene name was mentioned

L382-384 This seems like a very general conclusion to reach from a single case.

L452 - I couldn't understand this line

L583-585 It would be very useful if synteny data were presented for all the genes

L671 'elsewise' -> 'otherwise'?

Figure2A and Figure 3A dates don't match! Powell 1997 is rather an old reference.

Referee #2:

Riedelbauch and co-authors present thorough analyses of evolutionary dynamics of protein-protein interactions (PPIs) among the piRNA pathway proteins and general transcription factors in *Drosophilids*. The huge array of Y2H data is corroborated by the colocalization, mass-spec and ReLo experiments *in vivo*. These data allowed the authors identify three patterns of evolutionary turnover of PPIs and propose several hypotheses to explain what drives these changes. I support the publication of this work at EMBO Journal, given a reasonable effort is made to address a single major comment. Specifically, PPIs are an excellent proxy for the potential functional significance, yet an interaction may still be inconsequential for the biological function of either of the two proteins. If feasible, could authors use genetics to disrupt PPIs that are only found outside of *D. melanogaster*, to test if these PPIs are required for piRNA production and/or fertility? Alternatively, authors could introduce these interactions in *D. melanogaster* (with tethering tags) to test if such new interactions change the piRNA repertoire as one of the authors' hypotheses suggest. Such data would turn this rigorous work into a textbook case for studying evolutionary innovations with PPIs.

Referee #3:

In this study, comprehensive analyses of interactions among piRNA biogenesis factors in *Drosophila* were performed using different techniques, including high-quality Y2H. These analyses revealed that some interactions among piRNA biogenesis factors undergo frequent evolutionary changes. The paper demonstrates complex interactions across multiple *Drosophila* species in a clear and well-written manner. Also, the models proposed by their interaction data are very interesting. Meanwhile, this study presents only the interactions between proteins, leaving it unclear which of these interactions are "functional" in various *Drosophila* species (i.e., whether they could disrupt the piRNA pathway). This limitation makes it challenging to discuss whether the loss of interactions truly holds evolutionary significance. Specific comments are provided below.

1. While the interaction data propose interesting models, it is regrettable that these cannot be validated. For instance, would KO or KD of E(z) in *D. sim* or *D. ere* using CRISPR show an effect on piRNA production levels via small RNA-seq? Is the Boot-CtBP interaction essential in *D. sim*? For example, Boot IP in *D. mel* reveals that the interaction partners vary between the C- and N-termini (Fig4A and FigS6B), which again suggests that it would be helpful to have some data connecting the interactions to function.
2. The discussion suggests that the evolution of piRNA biogenesis factors in *Drosophila* is rapid, but compared to what? Without an understanding of the general rate of evolution, this argument may be difficult to evaluate. For example, is the mutation rate higher than in siRNA or miRNA pathway factors?
3. Regarding some factors, such as Kipferl, for which expression cannot be confirmed in certain *Drosophila* species: are these factors unnecessary for piRNA biogenesis in species where they are not expressed? Could other factors be compensating for their roles?
4. For the interactions among piRNA biogenesis factors reported in fruit flies that cannot be confirmed via Y2H, does this mean that such interactions are inherently difficult to observe with this method?
5. Regarding the data in Fig3H and FigS5D-F, while the co-localization of Rhino and GFP-Deadlock is observed, where does endogenous *D. melanogaster* Deadlock localize? Does it co-localize with GFP-Deadlock? If so, it would be difficult to determine whether the observed localization is due to endogenous Deadlock or GFP-Deadlock.
6. Why is the interaction between Boot and UAP56 observed in *D. vir* but not in *D. mel*? (Fig 2D) Could it be that a complex is required in *D. mel* but not in *D. vir*?
7. In Fig1B, why is there no blue dot visible for UAP56? Is it overlapping with the orange dot? The lack of an orange dot for Kipferl seems to be due to Kipferl's absence, which could be noted in the legend.

Dear editor and referees,

We thank all three referees for their detailed and constructive feedback on our work. While all
three reports are generally positive about our work, each has contributed valuable suggestions
for improvements. We are very grateful for these and have substantially improved the
manuscript accordingly.

We have addressed each of the referees' points in our revised manuscript, which includes added
synteny analysis, revised evolutionary analyses, new imaging data addressing the functional
implications of protein-protein interaction evolution, and many improvements to the text.

The following changes to figure panels have been implemented:

• Figure 1B: updated dN/dS and CODEML analyses (also related Appendix Figure S1)

• New figures: Appendix Figures S2-5 shows the requested synteny analysis

• New figure: Appendix Figure S7 now shows zoomed-in localization of N- and C-
terminally tagged *D. mel.* Bootlegger

• Figure 5K: Exchange of the representative ReLo image in the leftmost panel as this
image during source data preparation was found to have been wrongly placed (no
changes in conclusions as the correct image shows the same localization pattern).

• Several minor (cosmetic) figure changes and renaming to conform with EMBO J
guidelines

• Response-to-reviewer Figures R1-3 have been appended to this response as part of our
response to the comment from referee #2

Please find below our point-by-point response to each of the comments made by the reviewers.

We have copy-pasted the review comments below and added our response in green, bold
writing.

Best regards,

Peter Andersen (on behalf of all authors)

-----

Referee #1:

This manuscript presents an analysis of protein-protein interaction among components of the piRNA
biogenesis pathway across 5 species of Drosophila. The authors identify cases of conserved
interaction (proteins interact within and among species), coevolved interaction (proteins interact
within but not among species), and changes in pathway 'wiring' (interactions vary among species).
They discuss these findings in the light of ongoing hypotheses about the, unusually fast, evolution of
piRNA pathway components in Drosophila.

This paper represents an large and impressive piece of work. The logic seems sound, and the findings
will be of great interest to anyone working on piRNAs or pathway evolution more generally. The
work is well motivated, and the manuscript written and presented very well.

Any major concerns will, I think, arise from the reliability of Y2H screening - which I am not well
placed to assess critically. However, it does appear that they have taken as robust an approach as
possible, and because many of the claims are further supported using CoIP/Mass Spec. and/or co-
localisation based on microscopy, my own feeling is that the overall take-home messages are robust
and interesting, even if some interactions (and especially missing interactions) may prove to be
spurious.

Below, I list some individual points below (most trivial), in line order

L38 adaptive evolution is a response to positive selection, and therefore a marker of it - but the terms
are not synonymous.

**Indeed this was not a precise use of the terms. We have rephrased the wording for improved
precision (Lines #36-37 in the revised manuscript).**

L48 emphasise that few /experimental/ studies have done so

**We have added this emphasis on the lack of experimental studies to the mentioned part (Line
#46 in the revised manuscript).**

L93, L100, 159 and elsewhere - many of the in-text citations are mis-formatted with author initials.

This is a bug in Zotero.

**Thanks for noticing this. It is apparently also a bug in Paperpile, which we use. We have fixed**
**the issue with citation misformatting in the revised manuscript by switching to the EMBO Press**
**citation style in our Paperpile reference manager.**

Introduction: Overall is a bit wordy - I think it could be shortened by 20% with no loss of clarity.

**We agree the introduction was a bit long. We have now shortened it in a several places in the**
**revised manuscript.**

L128-153 and 560-568. This section on evolutionary rates is a bit weak.

**We thank the reviewer for the detailed input to improve these analyses. We have sought to**
**address each point (see specific info below by points (i) to (iii)).**

Most notably, (i) K_a/K_s between Dper and Dpse will be very noisy and dominated by polymorphism,
as K_s is only ~3%. Such an analysis should aim for species between $K_s=10\%$ and 30% divergent.

**To address the issue of comparing too closely related species, we have exchanged the Dper vs**
**Dpse analyses with analyses of *D. erecta* vs *D. yakuba*, for which the divergence is in the**
**suggested range. Figure 1B in the revised manuscript has been updated with the new data, and**
**source data has been added to provide the plot's underlying data.**

(ii) The models are insufficiently described in the results, and in any case M7 vs M8 is not good, and
best practice would be M8A vs. M8.

**To address possible issues with using the less stringent test of comparing models M7 and M8, we**
**have additionally included the stricter tests comparing the M8 and M8a models in our revised**
**analyses. With the more stringent test, four out of five positively selected genes from the M7 vs**
**M8 analyses are also found to be under positive selection when comparing the M8 and M8a**
**models, with deadlock (del) only being identified in the less stringent test. The added analyses,**
**therefore, confirm and substantiate our initial conclusions. We have updated Figure 1B and**
**Figure S1 (= new Appendix Figure S1) and the connected text of the results section (Lines #142-**
**146 in the revised manuscript) to reflect this. Furthermore, Source Data related to Figure 1B**
**has been added to provide the underlying PAML data. The relevant methods section has been**
**updated to reflect the new analyses (Lines #545-555 in the revised manuscript).**

(iii) I am very concerned that, because these proteins are so divergent, the inference from PAML
models will be spurious due to misalignment (AS noted in Figure S1 legend). How good do the
alignments look in regions that have inferred selection? Best Practice might be to use BAliPhy
(<https://doi.org/10.1093/bioinformatics/btab129> and <https://doi.org/10.1093/molbev/msu174>)

**It is indeed important to ensure that the alignments analyzed by PAML are solid. We have,**
**therefore, also taken great care to ensure this for our analyses. To provide this information to**
**the readers, we now include sequence alignments of all the analyzed gene segments as**
**supplementary Source Data files (attached here as Figure_1B_SD_Seq_align_for_PAML.zip).**

L193-197 I am not equipped to assess whether the 'multimeric complexes' argument is acceptable, or
a case of special pleading.

**Referee #3 phrased a question along the same lines (see Ref #3, comment 4 below + our**
**response). Briefly, the key interaction partners of Bootlegger and Trf2 are variants of the well-**
**characterized hetero-dimeric protein complexes, Nxf1–Nxt1 (PMID: 11780633) and TFIIA**
**(PMID: 17560669, 10570139), respectively. It is, therefore, very likely that Bootlegger**
**interaction with Nxf3 requires that Nxf3 binds its' interaction partner Nxt1 to form a Nxf3–**
**Nxt1 complex. Likewise, Trf2 interaction with Moonshiner or TFIIA-S, likely requires the**
**formation of a TFIIA-like complex by Moonshiner and TFIIA-S. We have now revised the**
**results text to clarify this inherent limitation of Y2H in detecting protein-protein interactions**
**that depend on forming multimeric protein complexes (lines #190-198 in the revised**
**manuscript).**

L233-239. I am dubious about the robustness of the claim "... at a similar high frequency ..." In
general, I would not accept such a claim without an explicit statistical test. However, after several
minutes thought I cannot formulate one, so perhaps the authors could not either.

**Indeed, we have also discussed this phrasing extensively. Despite the massive scale of our Y2H**
**screen, the number of species comparisons is still limited by the exponential scaling of the**
**number of Y2H tests needed for each extra species added. Future screens, likely using new**
**methodology or focusing on very few protein interactions, could provide sufficiently high intra-**
**and inter-species comparisons to allow statistical testing. To reduce the risk of misinterpreting**
**the statement as backed by statistical tests, we have now changed the phrasing to omit the word**
**'similarly' and instead of 'frequency' use 'fraction', which is also better aligned with the**
**wording in Figure 3B-F (Lines #234-240 in the revised manuscript).**

L250-252 "In sum ... are characterized by either conserved, co-evolving, or species-restricted
interactions" - does this mean "anything can happen" ?

**Indeed and this is contrary to the expected conserved interactions in essential protein**
**interaction networks. We have added a sentence to stress this (Lines #251-254 in the revised**
**manuscript).**

L286-296 I found this section hard to follow, as the species of origin was not always explicit in each
case that a Dmel gene name was mentioned

**Thanks for pointing out this unclear phrasing. We have edited the section to improve clarity by**
**specifying the relevant species names (Lines #290-303 in the revised manuscript).**

L382-384 This seems like a very general conclusion to reach from a single case.

**We have restricted the conclusion by rephrasing the conclusion sentence to better reflect our**
**proof-of-principle biochemical characterization focused on a single interaction in the related**
**figure (lines #387-388 in the revised manuscript).**

L452 - I couldn't understand this line

**We have rephrased the sentence for clarity (lines #455-458 in the revised manuscript).**

L583-585 It would be very useful if synteny data were presented for all the genes

**Synteny data for all the investigated genes is now included as Appendix Figure S2-S5 in the**
**revised manuscript (referred to in lines #170 (Results) and #572 (Methods) in the revised**
**manuscript).**

L671 'elsewise' -> 'otherwise'?

**This has now been corrected (line #658 in the revised manuscript).**

Figure2A and Figure 3A dates don't match! Powell 1997 is rather an old reference.

**Thanks for noticing this! During manuscript preparation, we shifted from an older dating**
**estimate but missed to correct Figure 2A. We have now corrected the dating in Figure 2A in the**
**revised manuscript so it matches Figure 3A. Furthermore, we now reference two central works**
**on estimating the divergence dates: Russo 1995 and Obbard 2012 (line #231 of the revised**
**manuscript).**

Referee #2:

Riedelbauch and co-authors present thorough analyses of evolutionary dynamics of protein-protein
interactions (PPIs) among the piRNA pathway proteins and general transcription factors in
Drosophilids. The huge array of Y2H data is corroborated by the colocalization, mass-spec and ReLo
experiments in vivo. These data allowed the authors identify three patterns of evolutionary turnover of
PPIs and propose several hypotheses to explain what drives these changes. I support the publication of
this work at EMBO Journal, given a reasonable effort is made to address a single major comment.

Specifically, PPIs are an excellent proxy for the potential functional significance, yet an interaction
may still be inconsequential for the biological function of either of the two proteins. If feasible, could
authors use genetics to disrupt PPIs that are only found outside of *D. melanogaster*, to test if these
PPIs are required for piRNA production and/or fertility? Alternatively, authors could introduce these
interactions in *D. melanogaster* (with tethering tags) to test if such new interactions change the piRNA
repertoire as one of the authors' hypotheses suggest. Such data would turn this rigorous work into a
textbook case for studying evolutionary innovations with PPIs.

**We are very thankful for the referees' positive evaluation of our work. Furthermore, we**
**much agree that *in vivo* genetic tests of the protein-protein interaction innovations within the**
**relevant *Drosophila* lineages are essential to probe the functional implications of piRNA**
**pathway PPI evolution with the highest possible rigor. As we outline below, such *in vivo* tests,**
**however, will require the generation of several genome-engineered lines in species where the**
**needed tools are not well established. Below our outline of why this work constitutes a work**
**package that will be a paper of its own, we describe alternative imaging-based efforts we have**
**made to shed light on the possible functional impact of the identified piRNA pathway PPI**
**evolution.**

**Regarding the suggestion to disrupt PPIs that are only found outside *D. melanogaster*,**
**such as the Bootlegger–CtBP interaction in *D. simulans*, while this experiment is ideal for test**
**the functional impact of PPI innovation, we consider it to be beyond the scope of the present**
**manuscript for three reasons.**

**First, making the genetically engineered *D. simulans* fly lines is not yet standardized,**
**and it would likely take around half a year before we have the needed *D. sim.* Bootlegger stocks.**
**<https://www.biorxiv.org/content/10.1101/2023.06.17.545412v2.full> describes CRISPR-Cas9**
**HDR-based genome editing as recently been described for *D. simulans* that we are currently**
**testing to implement in our lab for future work in this direction.**

**Second, to rigorously test the effect of the interaction site mutants from Figure 5J in our**
**paper, we would first need to validate that the PPIs also take place *in vivo* in *D. simulans*. For**
**this, we would need to endogenously add tags to the wildtype and the mutated *D. simulans* Boot**
**gene and following assay if adding those tags do not affect Boot function (TE expression by RT-**
**qPCR, piRNA production by small RNA-seq). *In vivo*, tagged Boot interactions would then be**
**assayed by coIP-MS before wildtype vs. CtBP-PPI mutant *D. simulans* Boot would be**
**functionally compared by their ability to support CtBP function at germline piRNA source loci**
**(assayed by detection of transposon-encoded transcription start site activity by, e.g., CAGE-seq**
**or STRIPE-seq). In other words, answering this question rigorously requires not just one quick**
**experiment but a whole new project.**

**Third, it remains possible that testing a single PPI will not give clear answers to the**
**broader question of PPI innovation function. For example, mutating the *D. simulans* Boot–CtBP**

interaction may be functionally compensated by alternative CtBP recruitment routes. If such
fail-safe CtBP recruitment exists, we would not see an effect of mutating the CtBP interaction
surface in Boot. However, this would not necessarily mean the interaction is not functional or
important.

We have also considered the referees' suggestion to introduce the interactions in *D.*
*melanogaster*. However, we have not been able to devise a meaningful way to do this. For
example, if we would tether CtBP to endogenous Bootlegger in *D. melanogaster* using e.g. GFP
and anti-GFP nanobody tags fused to the proteins, CtBP (endogenous and tagged transgenic)
would still get recruited via the unknown endogenous recruitment route in *D. melanogaster*. It,
therefore, seems unlikely that we would observe an easily interpretable result from such a test.
Such complications of functional tests in mixed species origin interactions highlight the strength
and importance of the first approach suggested by the referee, where the interactions are tested
in their native species and genetic environment. While this, for reasons outlined above, will be
part of our future work, genetic engineering and perturbation in non-melanogaster *Drosophila*
will be required to rigorously investigate the *in vivo* function of non-melanogaster *Drosophila*
protein biochemistry. To reflect these points, we have added a text to the discussion,
emphasizing the limitations of the present study and highlighting the importance of future
studies addressing the function of rapid protein-protein interaction evolution (lines #528-534 in
the revised manuscript).

Still, in an effort to characterize the functional impact of the PPI innovation in non-
melanogaster *Drosophila* piRNA pathways, we have performed immunofluorescence (IF)
imaging of endogenous CtBP, Trf2, and E(z) in *D. melanogaster*, *D. simulans*, and *D. erecta* to
investigate if these proteins accumulate at Rhino-dependent piRNA source loci to different
extents, possibly due to species-specific PPIs. The results are appended here as Response-to-
reviews Figures R1-3 (see bottom of the document).

First, aiming to characterize protein accumulation at Rhino-dependent germline piRNA
source loci in *D. mel.*, *D. sim.*, and *D. ere.*, we first screened available antibodies to find a reagent
that would allow us to detect piRNA source locus foci in ovary nurse cells in the three species.
For this, we performed IF staining on ovaries of each of the three species using eight different
mouse monoclonal antibodies previously raised against. The results (Fig. R1) showed that
mouse anti-UAP56 antibody can detect Rhino-like nuclear foci in *D. sim.*, and *D. ere.* (Fig. R1,
orange highlights). This result allows the detection of piRNA source loci by IF in all three
species. Furthermore, the failure of all the other antibodies raised against *D. mel.* piRNA
pathway proteins in IF on *D. sim.* and *D. ere.* ovaries is consistent with the rapid evolution of
protein surface epitopes in these proteins.

Next, we validated that the mouse anti-UAP56 antibody is a good proxy for Rhino foci
by co-staining with a rabbit anti-Rhino antibody that works in *D. sim.* but not in *D. ere.* (Fig.

**R2, leftmost column). The results showed that many UAP56 nuclear foci colocalize with Rhino.**
**In addition to its broadly nuclear distribution, anti-UAP56 IF, therefore, marks Rhino-**
**dependent piRNA source loci in *D. sim.* similar to in *D. mel.***

**To assay if PPI evolution has resulted in differential protein accumulation at piRNA**
**source loci, we then performed co-staining of mouse anti-UAP56 with rabbit antibodies**
**targeting multiple transcription factors (Trf2, CtBP, E(z), and Su(z)12) that our study**
**implicated in species-specific piRNA pathway PPIs and included Tbp as a negative control. The**
**antibodies were kindly provided by the J Kadonaga (Trf2, Tbp), J Müller (E(z) and Su(z)12),**
**and D Arnosti (CtBP) labs. The results show that most of the tested proteins in all three species**
**do not show substantial accumulation at UAP56 foci, consistent with previous observations for**
**Trf2 in *D. mel.* (Andersen, Nature, 2017; PMID: 28847004). Notably, the lack of differential**
**protein accumulation cannot be taken to conclude that PPI innovation has not had a functional**
**impact, as it is possible that PPIs evolve to maintain similar levels of transcription factor**
**recruitment via ever-changing PPI interfaces.**

**However, we did observe one transcription factor, CtBP, for which IF in *D. ere.* shows**
**focal accumulation overlapping with UAP56 foci (Fig. R2 and highlighted in Fig. R3). This**
**accumulation in *D. ere.* is consistent with the observed strong interaction between *D. ere.* Del**
**and CtBP by Y2H and AP-MS (Figs. 5C, EV2, and EV4C) and the detection of a likely CtBP-**
**interaction motif specifically in *D. ere.* Del (Fig. EV5E). We do note that this preliminary result**
**has some important limitations that relate to our general point in response to Referee #2: to**
**make a solid statement about this putative effect of PPI innovation, more work is needed,**
**warranting a separate study. Specifically, we have not yet directly demonstrated that the UAP56**
**foci that colocalize with CtBP are bound by endogenous *D. ere.* Rhino. Furthermore, it would**
**be essential to validate our IF-based observation by complementary approaches such as CtBP**
**ChIP-seq.**

**In sum, our IF-based investigation of transcription factor recruitment in three**
***Drosophila* species is consistent with the rapid gene evolution and PPI innovation presented in**
**our paper. Still, we feel that these efforts are rather the beginning of a new investigation and**
**have (for now) not added the newly generated IF data to the revised manuscript.**

Referee #3:

In this study, comprehensive analyses of interactions among piRNA biogenesis factors in *Drosophila*
were performed using different techniques, including high-quality Y2H. These analyses revealed that
some interactions among piRNA biogenesis factors undergo frequent evolutionary changes. The paper
demonstrates complex interactions across multiple *Drosophila* species in a clear and well-written

manner. Also, the models proposed by their interaction data are very interesting. Meanwhile, this
study presents only the interactions between proteins, leaving it unclear which of these interactions
are "functional" in various *Drosophila* species (i.e., whether they could disrupt the piRNA pathway).
This limitation makes it challenging to discuss whether the loss of interactions truly holds
evolutionary significance. Specific comments are provided below.

1. While the interaction data propose interesting models, it is regrettable that these cannot be
validated. For instance, would KO or KD of E(z) in *D. sim* or *D. ere* using CRISPR show an effect on
piRNA production levels via small RNA-seq?

**Genetic tools for gene perturbation in non-melanogaster *Drosophila* species remain**
**rudimentary (see also our above reply to referee #2). However, a recent preprint (Akkouche et**
**al. 2024) has indeed described the role of E(z) in piRNA production in *D. melanogaster*. In their**
**paper, Akkouche and colleagues show that ovary germline knock-down (KD) of E(z) results in**
**upregulation of transposons due to loss of piRNA production from several major germline**
**piRNA clusters and stand-alone transposon insertions. We now emphasize this finding in the**
**Discussion section (lines #478-481 in the revised manuscript).**

Is the Boot-CtBP interaction essential in *D. sim*?

**Please see our reply to Referee #2, who phrased a very similar question regarding the functional**
**impact of this specific interaction.**

For example, Boot IP in *D. mel* reveals that the interaction partners vary between the C- and N-
termini (Fig4A and FigS6B), which again suggests that it would be helpful to have some data
connecting the interactions to function.

**The difference in interaction partners between C- and N-terminally tagged Bootlegger protein**
**lies in that N-terminally tagged Bootlegger does not interact with the other export components**
**Nxf3 and UAP56 (Figure 4A and S6B (= new Figure EV4)). New, close analysis of our**
**Bootlegger localization imaging data now revealed that N-terminally tagged Bootlegger protein**
**is indeed functionally impaired, as evidenced by the absence of the cytoplasmic perinuclear foci**
**observed for C-terminally tagged Bootlegger (new Appendix Figure S7). We now emphasize this**
**functional impact in the revised Results section (lines #297-298 in the revised manuscript).**

2. The discussion suggests that the evolution of piRNA biogenesis factors in *Drosophila* is rapid, but
compared to what? Without an understanding of the general rate of evolution, this argument may be
difficult to evaluate. For example, is the mutation rate higher than in siRNA or miRNA pathway
factors?

**This comment points to an important context of rapidly evolving genes that was partially**
**missing in our submitted manuscript. In fact, a comparative analysis of genes involved in**
**siRNA, miRNA, and piRNA pathways in *Drosophila* shows that genes involved in miRNA**
**biology evolve slowly. In contrast, genes of the siRNA and piRNA pathways evolve rapidly**
**(Figure 4 in Obbard et al. 2009, PMID: 18926973). We now refer to this work in the relevant**
**Discussion section (lines #410-412 in the revised manuscript).**

3. Regarding some factors, such as Kipferl, for which expression cannot be confirmed in certain
*Drosophila* species: are these factors unnecessary for piRNA biogenesis in species where they are not
expressed? Could other factors be compensating for their roles?

**Kipferl has been identified only in species of the melanogaster subgroup (Baumgartner et al.**
**2022, PMID: 36193674), which includes *D. mel.*, *D. sim.*, and *D. ere.* and further species with a**
**last common ancestor around 6 million years. Kipferl belongs to the ZAD zinc finger (ZF)**
**genes, which includes over 90 genes in *D. melanogaster*. Analogous to the KRAB-ZF pathway in**
**mammals, where different KRAB-ZF proteins guide the KRAB1 silencing factor to various loci,**
**it is therefore possible that, e.g., other ZAD-ZF genes specify Rhino-bound loci in various**
***Drosophila* species. The second possibility outlined by the referee, however, also has precedence**
**in the literature, where e.g. Chary and Hayashi 2023 (PMID: 37279192) describe how in *D.***
***eugralis*, a species of the *melanogaster* subgroup, lack two core piRNA pathway proteins**
**describe in *D. mel.* , Yb and Ago3. The *D. eugralis* piRNA pathway remains competent in**
**transposon silencing but has changed to function without ping-pong amplification and Yb-**
**specification of somatic piRNA precursors. Both possibilities noted by the referee are, therefore,**
**possible. To keep the manuscript concise, we have not included this discussion in our revision**
**but are open to doing so if the referees and/or editor feel this would improve the paper.**

4. For the interactions among piRNA biogenesis factors reported in fruit flies that cannot be
confirmed via Y2H, does this mean that such interactions are inherently difficult to observe with this
method?

**Indeed, we believe this to be the case. The inability of Y2H to recapitulate interactions that**
**require more than two proteins is a known characteristic of the methodology. While endogenous**
**yeast proteins, in principle, could support complex formation with highly conserved *Drosophila***
**proteins, most of the investigated genes are absent in yeast.**

**For example, co-IP/MS analyses show a strong interaction between Trf2, TFIIA-S, and**
**Moonshiner, likely forming a stable biochemical complex (Andersen, Nature, 2017; PMID:**
**28847004). By Y2H, we detect TFIIA-S–Moonshiner interaction, confirming that these proteins**
**may form a hetero-dimeric TFIIA-like complex similar to canonical TFIIA-L and TFIIA-S**
**(Høiby, Biochim Biophys Acta, 2007; PMID: 17560669). However, we did not observe any**

interactions with Trf2, indicating that this interaction requires the formation of the TFIIA-S-
Moonshiner TFIIA-like complex (Figure 2C-D).

Similarly, well-established biochemical interactions (coIP/MS) between Bootlegger and Nxf3
were also not recapitulated by Y2H. This is likely because Nxf3 *in vivo* forms a stable hetero-
dimeric complex with Nxt1 (ElMaghraby, 2019, Cell; PMID: 31398345 – similarly to canonical
Nxf1–Nxt1, Herold, 2001, RNA; PMID: 11780633). This Nxf3–Nxt1 complex formation is likely
required for interaction with Bootlegger, resulting in its absence in the Y2H-based binary
interaction tests (Figure 2C-D).

We have now revised the results text to clarify this inherent limitation of Y2H in detecting
protein-protein interactions that depend on forming multimeric protein complexes (lines #190-
198 in the revised manuscript).

5. Regarding the data in Fig3H and FigS5D-F, while the co-localization of Rhino and GFP-Deadlock
is observed, where does endogenous *D. melanogaster* Deadlock localize? Does it co-localize with
GFP-Deadlock? If so, it would be difficult to determine whether the observed localization is due to
endogenous Deadlock or GFP-Deadlock.

**Rhino, Deadlock, and Cuff localization to nuclear foci depend on the expression of each of the**
**proteins in *D. melanogaster* (Figure S4, Mohn 2014; PMID: 24906153). Furthermore, *D.***
***melanogaster* Rhino will form strong foci if *D. melanogaster* but not *D. simulans* Deadlock is**
**expressed (Figure 6D, Parhad 2017; PMID: 28919205). Given the presence of clear Rhino foci in**
**each GFP-Deadlock line, we infer that endogenous Deadlock must co-localize with endogenous**
**Rhino. Hence, the localization of GFP-Deadlock indeed relies on endogenous Deadlock for the**
**non-*melanogaster* species. However, our mass spectrometry results show that while *D. sim.* and**
***D. vir.* Deadlock prominently co-immunoprecipitate endogenous Rhino, no endogenous**
**Deadlock protein is observed in these *Ips* (Fig. EV4A). This suggests that the co-localization of**
***D. sim.* and *D. vir.* Deadlock with endogenous Rhino is due to a direct interaction, as also**
**supported by our Y2H data (Fig. EV2). To clarify the role of endogenous Deadlock in the**
**formation of Rhino foci, we have mentioned this in the results section (lines #268-270 in the**
**revised manuscript).**

6. Why is the interaction between Boot and UAP56 observed in *D. vir* but not in *D. mel*? (Fig 2D)
Could it be that a complex is required in *D. mel* but not in *D. vir*?

**We agree that it is indeed possible that the UAP56–Boot interaction in *D. vir.* has evolved to be**
**less dependent on complex formation with other interaction partners, such as Nxf3–Nxt1 and**
**the THO complex (see also our detailed reply to your comment #4 regarding Y2H and**
**multimeric complexes). We, however, also noted that the Y2H replication score for *D.vir.***
**UAP56 with *D.vir.* Boot is relatively low (0.8), suggesting the interaction is not very robust. To**

**avoid giving this putative interaction more focus than it can justify, we have, for now, opted not**
**to put specific emphasis of it in the revised manuscript.**

7. In Fig1B, why is there no blue dot visible for UAP56? Is it overlapping with the orange dot? The
lack of an orange dot for Kipferl seems to be due to Kipferl's absence, which could be noted in the
legend.

**The blue UAP56 dot was indeed hidden behind the orange dot in the previous version of Figure**
**1B – thanks for pointing this out. In the revised version of Figure 1B, we have shifted the two**
**UAP56 dots horizontally slightly apart to make them both visible. Furthermore, since we**
**exchanged the *D. per.* Vs. *D. pse.* comparison with one comparing the more diverged species *D.***
***ere.* and *D. yak.* (see referee #2 comment asking for this), Kipferl is now included in both**
**comparison analyses as it is present in all the involved species.**

Response-to-reviews Figure R1

Response-to-reviews Figure R1. Testing *D. melanogaster* antibodies in *D. simulans* and *D. erecta*

Confocal microscopy images showing whole ovarioles from ovaries dissected from the species noted on the top. Ovaries were stained by immunofluorescence (IF) using various mouse monoclonal (m) or rabbit (r) anti-Rhino antibodies raised against *D. mel.* piRNA pathway-related proteins. Orange boxes highlight IF stainings that give clear nuclear signal, which in *D. mel.* overlap Rhino foci from (r) IF. DNA was visualized by DAPI staining.

Response-to-reviews Figure R2

Response-to-reviews Figure R2. Investigating enrichment of transcription factors at Rhino foci

Confocal microscopy images showing single ovary nurse cell nuclei from ovaries dissected from the species noted to the left. Ovaries were stained by immunofluorescence (IF) using mouse monoclonal anti-UAP56 antibodies to mark piRNA source loci and various rabbit antibodies raised against *D. mel.* piRNA pathway-related proteins. Orange boxes highlight IF stainings that give nuclear signal that overlap UAP56 IF foci. DNA was visualized by DAPI staining.

Response-to-reviews Figure R3

Response-to-reviews Figure R3. Investigating enrichment of transcription factors at Rhino foci

Confocal microscopy images as described in Figure R2 but showing two 3x enlarged *D. erecta* nurse cell nuclei to better visualize the overlapping focal signal.

Dear Dr. Andersen,

Thank you for submitting your manuscript for consideration by the EMBO Journal. It has now been seen by the three original referees from the previous round whose comments are enclosed. As you will see, all three referees express interest in your manuscript and support its favour of publication.

Given the referees' positive recommendations, I would like to invite you to submit a revised version of the manuscript, addressing the very minor comments by Referee #1 and fixing mainly technical comments by our editorial assistant that you will find below.

When preparing your letter of response to the comments, please bear in mind that this will form part of the Review Process File, and will therefore be available online to the community. You do not need to include your responses to the Assistant's comments, but you need to follow all of them thoroughly and incorporate the edits into your revision. For more details on our Transparent Editorial Process, please visit our website: <https://www.embopress.org/page/journal/14602075/authorguide#transparentprocess>

We generally allow three months as standard revision time and this is what our system will automatically allocate in your case. Yet, I expect that you could submit your revision much sooner in light of the fact these are all relatively small tasks.

Thank you for the opportunity to consider your work for publication. I look forward to your revision.

Yours sincerely,

Yehu Moran
Academic Editor
The EMBO Journal

General instructions for preparing your revised manuscript:

We realize that it is difficult to revise to a specific deadline. In the interest of protecting the conceptual advance provided by the work, we recommend a revision within 3 months (13th Jun 2025). Please discuss the revision progress ahead of this time with

the editor if you require more time to complete the revisions.

Editorial Assistant comments:

MANUSCRIPT FORMAT: Please make sure to upload a .docx file, with no figures, no track changes.

FUNDING INFO: The following are missing in our system: AIAS Marie Curie CO-FUND Fellowship; Field of Excellence BioHealth - University of Graz. Please make to insert all of them in our system.

Conflict of Interest: title needs renaming to "DISCLOSURE AND COMPETING INTERESTS STATEMENT".

AC/CRedit: section needs to be removed from the text file and filled only in our online system.

FIGURE CALLOUTS: all callouts should be listed sequentially; callout for Table S2 should be corrected to Appendix Table S2; missing callout for Fig. 4C,

Checklist: missing. Please provide.

Figures in separate files: Figures should be uploaded to our system as individual Figure files.

APPENDIX 1 FILE WITH Table of Contents: Appendix file needs to be in PDF format; title page should be: "Appendix for <>" and Table of Contents with the page numbers for the listed items. You can find plenty of examples in previously published manuscripts.

R&T TABLE: in ms file; should be uploaded as an individual file using the template from our GTA.

SOURCE DATA: Source data files need to be saved in a scheme one figure/folder and then uploaded as .zip files. E.g. all the Source data files for figure 1 need to be saved in a single folder and this needs to be zipped and then uploaded as "SD figure 1.zip" file. For EV and/or appendix figures, ZIP together all source data. Completed SD checklist should be uploaded as Related Manuscript File.

SYNOPSIS IMAGE: missing. Please provide.

SYNOPSIS TEXT: missing. Please provide.

FIGURE CHECK:

Figure reuse between Figure 3H and Figure EV3 D&E. Figure EV3 legend says (related to Figure 3) The cell reuse needs to be detailed. Also is the reuse necessary or does it add to the paper? If yes, please keep it but explicitly mention the reuse.

Figure reuse between Figure EV3F and Appendix Figure S7. Appendix Figure S7 legend states data related to Figure 4. The cell reuse between Figure EV3F needs to be detailed in the legend (see above).

- Figure Legends (main + EV): Please note that the error bars are not defined in the legends of figures 4C, D, G, H, I. Please define them.

- Section order for the manuscript should be corrected: Title page - Abstract & Keywords - Introduction - Results - Discussion - Methods - Data Availability - Acknowledgements - Disclosure and Competing Interests Statement - References - Figure Legends - Table(s) - Expanded View Figure Legends.

- Subject categories missing, please provide.

Referee #1:

I have no further comments. I think this is a great paper, and I am very happy with the authors revisions and responses to my earlier comments - but I do not feel sufficiently qualified to comment on their responses to the other reviewers.

Two very minor points.

[1] A very recent paper emphasises that the rapid evolution seen in the piRNA pathway of the melanogaster group is not universal (<https://doi.org/10.1093/g3journal/jkaf017>), and this might be worth commenting on. Especially with regard to the higher divergence species virilis.

[2] The degrees of freedom for the M8 / M8a test is not straightforward - "From standard theory (Chernoff 1954), it follows that the log-likelihood ratio statistic is asymptotically distributed as a 50:50 mixture of a point mass at zero and a chi-distribution." <https://doi.org/10.1093/oxfordjournals.molbev.a004233>. However, the approach used here is conservative (if under-powered) so I think it is not a problem.

Referee #2:

The authors have addressed my concerns by performing additional experiments and adjusting the text.

Referee #3:

In this study, comprehensive analyses of interactions among piRNA biogenesis factors in *Drosophila* were conducted using various techniques, including high-quality Y2H. These analyses revealed that some interactions among piRNA biogenesis factors undergo frequent evolutionary changes.

While solid experimental validation of PPI, particularly in *Drosophila* species, remains somewhat lacking in the revised manuscript, the revisions have been made in a logical and clear manner. The paper demonstrates complex interactions across multiple *Drosophila* species from an evolutionary perspective. Given this, I believe it is suitable for acceptance in its current form.

Aarhus, March 22, 2025

Dear referees,

We are very happy with the positive comments from all three referees and have now addressed the referees' additional comments. The manuscript and figures have undergone the following minor changes:

- (i) We now cite the paper recommended by referee #1 with the following sentence:
"Of note, it was recently reported that piRNA pathway genes also show rapid evolution signatures in *D. ananassae* and *D. willistoni*, although to a lesser extent than *D. melanogaster*, suggesting that the driver(s) of rapid piRNA pathway evolution are dynamic (Blumenstiel *et al*, 2025)."
- (ii) As referee #1 correctly points out, our statistical analyses for positive selection using the M8-M8a model comparison was unnecessarily conservative as it was done with a setting of 2 degrees of freedom. We have updated the p values in Appendix Figure S1 to those based on analyses with a setting of 1 degree of freedom and updated the related text in the methods section to: "The degrees of freedom for each test were set to 2 for the M8-M7 comparison and 1 for the M8-M8a comparison". As expected, the change lowered all p value, but did not change the results in any substantive way.
- (iii) Proteins IDs for the Y2H experiments have been added as Table EV1 (formerly Table S2) (file: Table_EV1_Protein_IDS_Y2H.xlsx).
- (iv) Figure EV2 (Y2H replication score matrix) has been updated for increased readability (aesthetics only). The updated figure can be reproduced from the related Source Data.

Thanks to all three referees for their constructive input, which has helped improve our manuscript.

With our best wishes,

Peter Andersen
On behalf of all authors

Dear Dr. Andersen,

I am pleased to inform you that your manuscript has been accepted for publication in the EMBO Journal. Congratulations on an excellent academic work!

Yours sincerely,

Yehu Moran
Academic Editor
The EMBO Journal
